# Learning with Auxiliary Activation for Memory-Efficient Training

**Sunghyeon Woo, Dongsuk Jeon**
Seoul National University, Seoul, Korea
{wsh0917,djeon1}@snu.ac.kr

## Abstract

While deep learning has achieved great success in various fields, a large amount of memory is necessary to train deep neural networks, which hinders the development of massive state-of-the-art models. The reason is the conventional learning rule, backpropagation, should temporarily store input activations of all the layers in the network. To overcome this, recent studies suggested various memory-efficient implementations of backpropagation. However, those approaches incur computational overhead due to the recomputation of activations, slowing down neural network training. In this work, we propose a new learning rule which significantly reduces memory requirements while closely matching the performance of backpropagation. The algorithm combines auxiliary activation with output activation during forward propagation, while only auxiliary activation is used during backward propagation instead of actual input activation to reduce the amount of data to be temporarily stored. We mathematically show that our learning rule can reliably train the networks if the auxiliary activation satisfies certain conditions. Based on this observation, we suggest candidates of auxiliary activation that satisfy those conditions. Experimental results confirm that the proposed learning rule achieves competitive performance compared to backpropagation in various models such as ResNet, Transformer, BERT, ViT, and MLP-Mixer.

## 1 Introduction

Backpropagation (1) is an essential learning rule to train deep neural networks and has proven its outstanding performance in diverse models and tasks. In general, the wider and deeper the deep learning model, the better the training performance (2). However, increasing model size unavoidably requires larger memory in training hardware such as GPU (3; 4). This is because backpropagation needs to temporarily store input activations of all the layers in the network generated in forward propagation as they are later used to update weights in backward propagation. Consequently, state-of-the-art deep learning models require substantial memory resources due to the large amount of input activation to store. In order to train very deep models with limited hardware resources, the batch size may be reduced (3; 4) or many GPUs can be used in parallel (5; 6; 7; 8; 9; 10; 11). However, reducing the batch size causes a long training time, and the advantage of batch normalization (12) disappears. Also, training huge models such as GPT-3 (13) still requires expensive GPU clusters with thousands of GPUs and incurs high I/O costs even with parallelism.

Recently, a wide range of algorithms have been proposed to alleviate this memory requirement. For instance, a new optimizer (14; 15) or neural network architectures (16; 17; 18; 19; 20; 21) have been suggested to reduce the memory requirements. Gradient checkpointing (22; 23; 24; 25; 26) reduces memory space by only storing some of the input activation during forward propagation. Then, it restores the unsaved input activations through recomputation in backward propagation. In-place activated batch normalization (27) merges a batch normalization layer and a leaky ReLU layer and stores the output activation of the merged layer in forward propagation. In backward propagation, the layer input can be reconstructed for training because the leaky ReLU function is reversible. Similarly, RevNet (28), Momentum ResNet (29), and Reformer (30) employ reversible neural network architectures, which allow for calculating input activation from output activation in backward propagation. Gradient checkpointing and reversible network structures reduce training memory space because they partially store input activations (e.g., input activations of selected layers). However,

these methods incur additional computational overhead because the unstored input activations must be recomputed during backward propagation. Alternatively, algorithms to approximate activation have been suggested (31; 32; 33; 34; 35; 36; 37; 38), but they suffer from performance degradation or slow down training due to additional computations to quantize and dequantize activations. TinyTL (39) entirely avoids saving activations by updating only bias parameters while fixing weight parameters. However, it is only applicable to fine-tuning of a pre-trained model.

In this study, we propose a new learning rule, Auxiliary Activation Learning, which can significantly reduce memory requirements for training deep neural networks without sacrificing training speed. We first introduce the concept of auxiliary activation in the training process. Auxiliary activations are combined with output activations and become the input activation of the next layer when processing forward propagation, but only the auxiliary activations are temporarily stored instead of actual input activation for updating weights in backward propagation. To justify our algorithm, we prove that an alternate type of input activation could reliably train neural networks if auxiliary activation satisfies certain conditions. Then, we propose multiple candidates of auxiliary activations which meet this criterion. Experimental results demonstrate that the proposed algorithm not only succeeds in training ResNet models (40) on ImageNet (41) with similar performance to backpropagation, but is also suitable for training other neural network architectures such as Transformer (42), BERT (43), ViT (44), and MLP-Mixer (45).

## 2 AUXILIARY ACTIVATION LEARNING

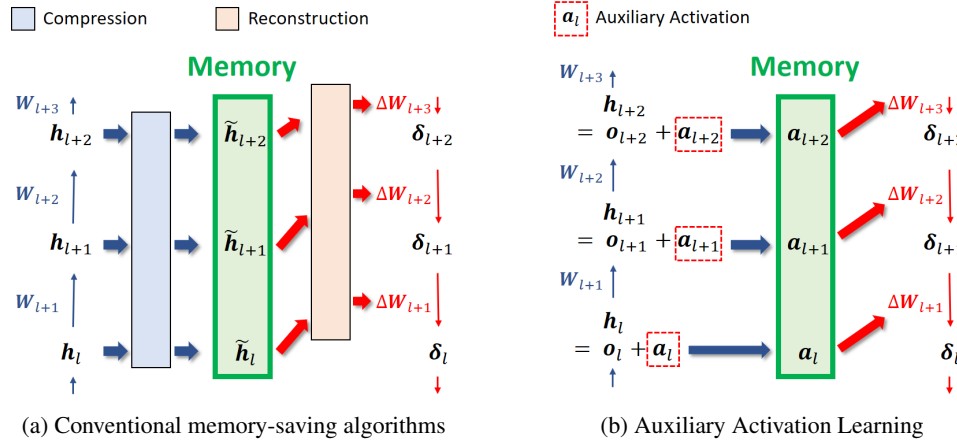

(a) Conventional memory-saving algorithms

(b) Auxiliary Activation Learning

Figure 1: Conventional memory-efficient training algorithms and Auxiliary Activation Learning.

### 2.1 MEMORY REQUIREMENTS OF BACKPROPAGATION

To understand why backpropagation uses a large memory space for training, we describe how it trains a deep neural network. The training process of backpropagation can be expressed by the equations below.

$$\boldsymbol{h}_{l+1} = \phi(\boldsymbol{W}_{l+1}\boldsymbol{h}_l + \boldsymbol{b}_{l+1}) \tag{1}$$

$$\boldsymbol{\delta}_l = \boldsymbol{W}_{l+1}^T\boldsymbol{\delta}_{l+1} \odot \phi^{'}(\boldsymbol{y}_l) \tag{2}$$

$$\Delta\boldsymbol{W}_{l+1} = -\eta\boldsymbol{\delta}_{l+1}\boldsymbol{h}_l^T \tag{3}$$

Here $\boldsymbol{h}_l$, $\boldsymbol{W}_{l+1}$, and $\boldsymbol{b}_{l+1}$ denote the input activation, weight, and bias of the hidden layer $l+1$, respectively. $\phi$, $\boldsymbol{\delta}$, and $\eta$ represent the nonlinear function, backpropagated error, and learning rate, respectively. $\boldsymbol{y}_l$ is the output of a linear or convolutional layer (i.e., $\boldsymbol{y}_l = \boldsymbol{W}_l\boldsymbol{h}_{l-1} + \boldsymbol{b}_l$), which becomes the input of the nonlinear function $\phi$. In forward propagation, the input propagates through the network following equation (1). During this process, the generated input activations are stored in memory. In backward propagation, the error is backpropagated through the network based on

equation (2). The weights are updated using the backpropagated errors and the activations stored in memory (equation (3)). Due to the time gap between forward and backward computations, all activations obtained in forward propagation must be temporarily stored in memory as they are later used for backward propagation. Therefore, as the model size increases, the amount of activation to be stored in memory increases accordingly. To alleviate this issue, gradient checkpointing (22; 23; 24; 25; 26) stores only some of the activations in memory during forward propagation, and the rest of the activations are restored during backpropagation by recomputing equation (1). Similarly, activation compression methods (31; 32; 33; 35; 36; 37) quantize activations before saving them in memory to reduce memory requirements, and they are dequantized during backward propagation. However, these methods incur computational overhead and increase training time due to additional processes during forward and backward propagation as depicted in Fig. 1a.

## 2.2 AUXILIARY ACTIVATION LEARNING: STORE AUXILIARY ACTIVATION INSTEAD OF EXACT ACTIVATION

For memory-efficient training with minimal overhead, here we propose a new learning rule, *Auxiliary Activation Learning*. First, we introduce the concept of auxiliary activation in the training process. Then, the algorithm trains the model following the equations below.

$$h_l = o_l + a_l \tag{4}$$
$$o_{l+1} = \phi(W_{l+1} \, h_l + b_{l+1}) \tag{5}$$
$$\delta_l = W_{l+1}^T \delta_{l+1} \odot \phi^{'}(y_l) \tag{6}$$
$$\Delta W_{l+1} = -\eta \delta_{l+1} \, a_l^T. \tag{7}$$

Here $a_l$ represents the newly introduced auxiliary activation of layer $l$. In forward propagation, each layer takes the sum of auxiliary activation $a_l$ and the previous layer's output activation $o_l$ as input activation $h_l$ (equation (4)). The generated input activation propagates through the layer following equation (5). In backward propagation, the error $\delta_{l+1}$ is backpropagated by equation (6). Finally, the weights are updated only using auxiliary activation $a_l$ instead of the actual input activation $h_l$ following equation (7). The learning process of our algorithm is depicted in Fig. 1b. Now the memory needs to store the auxiliary activations $a_l$, not real activations $h_l$. Consequently, if the auxiliary activation is reused (e.g., $a_l = a_{l+1}$) or the amount by auxiliary activation itself is small, the required memory space can be reduced accordingly. Furthermore, it does not require additional computations, avoiding training time increases.

The intuition behind only using auxiliary activation for weight updates is that the direction of a weight update obtained using auxiliary activation could be similar to that of backpropagation if we carefully choose auxiliary activation. More specifically, we use the sum of the auxiliary activation $a_l$ and the previous layer's output activation $o_l$ as the input activation $h_l$ (equation (4)). If we take an existing value (e.g., activation from a different layer) as the auxiliary activation, this can be seen as making an additional connection to the layer. If we update weights using backpropagation as in equation (3), the input activation $h_l$ generated in forward propagation will be used for updating weights: $\Delta W_{l+1}^{BP} = -\eta \delta_{l+1} \, h_l^T = -\eta \delta_{l+1} \, (o_l + a_l)^T = -\eta \delta_{l+1} o_l^T - \eta \delta_{l+1} a_l^T$. Consequently, the weight gradient obtained by using our algorithm can be expressed by $\Delta W_{l+1}^{AAL} = \Delta W_{l+1}^{BP} + \eta \delta_{l+1} o_l^T$. Therefore, it is expected that the weight update direction of our algorithm can be similar to that of backpropagation if the effect of the second term is ignorable, which will result in proper training of the neural network. In the next section, we prove that this assumption can be realized if we choose the auxiliary activation appropriately.

## 3 ANALYSIS OF AUXILIARY ACTIVATION LEARNING

### 3.1 HOW AND WHEN USING ALTERNATIVE ACTIVATION CAN TRAIN DEEP NEURAL NETWORKS

This section shows how learning with an alternative form of activation can make the loss of nonlinear networks converge to a minimum value if a specific condition is satisfied. For mathematical analysis, we consider a loss function $f(x)$ whose gradient is Lipschitz continuous. $h$ and $r$ are column vectors representing real activations and random alternative activations, respectively. $\delta$ is a column vector

of backpropagated errors, and $W$ is a flattened vector of weights. When weights are updated by backpropagation using exact activations $h$,

$$\Delta W^t = \nabla f(W^t) = \text{Vec}(\delta_1 h_0^T, \ldots, \delta_L h_{L-1}^T) \tag{8}$$

where $t$ denotes the current epoch. Contrarily, now we assume that the weights are updated using alternative activations $r_l$ as below, which represents a generalization of our algorithm.

$$\Delta W^t = \text{Vec}(\delta_1 r_0^T, \ldots, \delta_L r_{L-1}^T) \neq \nabla f(W^t) \tag{9}$$

$$W^{t+1} = W^t - \eta \Delta W^t \tag{10}$$

**Theorem 1.** *If $r_l^T(2h_l - r_l) > 0$ in all layers, the step size $\eta$ satisfies $0 < \eta \leq \frac{1}{L}$, and the gradient of the loss function $f(W)$ is Lipschitz continuous, then $f(W)$ converges to a minimum value.*

Theorem 1 can be proven by applying equations (9)-(10) to the quadratic upper bound. A detailed proof is provided in Appendix A. Because condition $0 < \eta \leq \frac{1}{L}$ can be easily satisfied if we use a small step size, $r_l^T(2h_l - r_l) \geq 0$ is the most important condition for successful learning that alternative activations should satisfy. We name this condition *Learning Criterion*. Furthermore, the larger the value of $r_l^T(2h_l - r_l)$, the better the network converges because the amount of loss reduction in each epoch ($K^n$ in Appendix A) increases. One interesting observation is that $r_l^T(2h_l - r_l) = 2r_l \cdot h_l - \|r_l\|^2 = -\|h_l - r_l\|^2 + \|h_l\|^2$ reaches its maximum when $r_l = h_l$, which translates to a conventional backpropagation algorithm. For convenience, we normalize the learning criterion inequality with respect to its maximum value $\|h_l\|^2$ and define a learning indicator as below.

$$Learning\ Indicator \equiv \frac{r_l^T(2h_l - r_l)}{\|h_l\|^2} \tag{11}$$

In summary, alternative activation can train the network if the learning indicator in equation (11) is positive. Moreover, the larger the learning indicator, the better the training performance. Finally, the training performance is maximized when the learning indicator is 1 ($r_l = h_l$, backpropagation).

## 3.2 Constructing auxiliary activations

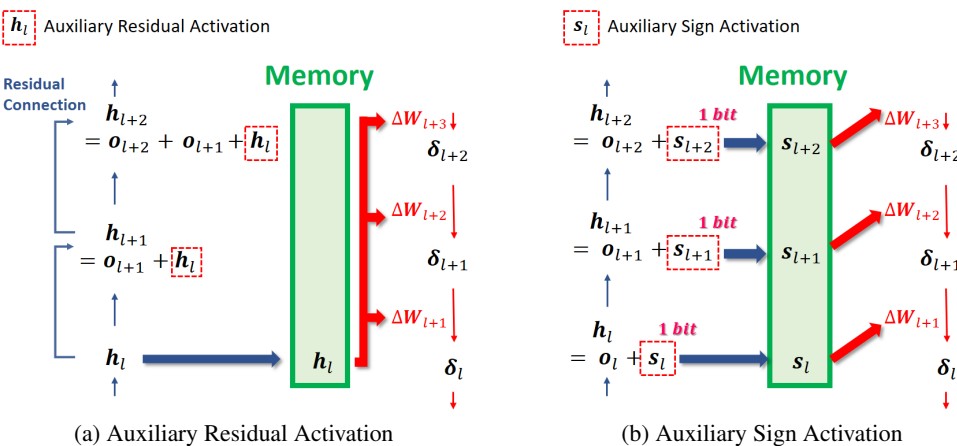

Figure 2: Two candidates of auxiliary activation.

Theorem 1 suggests that using an alternate type of activation in calculating weight gradient could still reliably train the neural network. However, in reality, it is cumbersome to meet the learning criterion if we use an arbitrary value as the alternate activation $r_l$ (see Appendix C). To address this, we modify the forward propagation path by adding auxiliary activation to the input activation as in equation (4) in the proposed Auxiliary Activation Learning. Then, the input activation $h_l = o_l + a_l$ and the alternate activation $r_l = a_l$ share a common term $a_l$, making it easier to meet the learning criterion. Also, if we employ some values already existing in the network as auxiliary activation $a_l$,

this modification can be regarded as introducing an additional connection in the layer. The learning indicator of Auxiliary Activation Learning is expressed by

$$Learning\ Indicator = \frac{\boldsymbol{r}_l^T(2\boldsymbol{h}_l - \boldsymbol{r}_l)}{\|\boldsymbol{h}_l\|^2} = \frac{\boldsymbol{a}_l^T(2\boldsymbol{o}_l + \boldsymbol{a}_l)}{\|\boldsymbol{h}_l\|^2} \tag{12}$$

by applying equation (4) to equation (11). For successful learning, the value of the learning indicator in equation (12) should be positive. To satisfy this learning criterion, we suggest two candidates of auxiliary activation as below.

**Auxiliary Residual Activation** First, we can use the previous layer's input activations as auxiliary activations and combine them with output activations, which is identical to having a residual connection. Then, the learning process is expressed by

$$\boldsymbol{o}_{l+1} = \phi(\boldsymbol{W}_{l+1}(\boldsymbol{o}_{l+1} + \boldsymbol{h}_l) + \boldsymbol{b}_{l+1}) \tag{13}$$

$$\Delta\boldsymbol{W}_{l+2} = -\eta\boldsymbol{\delta}_{l+2}\,\boldsymbol{h}_l^T \tag{14}$$

Using auxiliary residual activations can save training memory if we only store $\boldsymbol{h}_l$ and use it for updating $\boldsymbol{W}_{l+1}$ using backpropagation as well as for updating $\boldsymbol{W}_{l+2}$ and $\boldsymbol{W}_{l+3}$ using Auxiliary Activation Learning as depicted in Fig. 2a and equation (14). Contrarily, backpropagation needs to store $\boldsymbol{h}_l$, $\boldsymbol{h}_{l+1}$, and $\boldsymbol{h}_{l+2}$ for updating these three layers.

**Auxiliary Sign Activations** Another option is using the sign of the output activation as auxiliary activations. Then, the weight update process is represented by the equations below.

$$\boldsymbol{o}_{l+1} = \phi(\boldsymbol{W}_{l+1}(\boldsymbol{o}_l + \epsilon\text{sign}(\boldsymbol{o}_l)) + \boldsymbol{b}_{l+1}) \tag{15}$$

$$\Delta\boldsymbol{W}_{l+1} = -\eta\boldsymbol{\delta}_{l+1}\,\epsilon\text{sign}(\boldsymbol{o}_l)^T \tag{16}$$

The auxiliary sign activation ($\boldsymbol{s}_l = \epsilon\text{sign}(\boldsymbol{o}_l)$) can be guaranteed to satisfy the learning criterion. Because $2\boldsymbol{a}_l^T\boldsymbol{o}_l = 2\epsilon\text{sign}(\boldsymbol{o}_l)^T\boldsymbol{o}_l \geq 0$, the learning indicator in equation (12) is always positive. Using auxiliary sign activation can enable memory-efficient training by storing only a 1-bit sign of activation instead of actual input activation in high precision as shown in Fig. 2b. In equations (15) and (16), we multiply the auxiliary activation by a hyperparameter $\epsilon$ to make the two types of activation comparable in magnitude since the output of the sign function is -1 or 1.

## 4 Experimental Results

In experiments, we trained various deep neural networks using the proposed learning rule and compared its training performance to that of backpropagation. We observed if the selected auxiliary activations satisfied the learning criterion during training and also measured the amount of memory savings and training time. Our algorithm was used to train ResNet (40) as well as transformer and its variants (42; 43; 44; 45), which are widely used architectures in vision and NLP-related applications.

### 4.1 Training ResNet

We applied the proposed Auxiliary Activation Learning to ResNet training using two types of auxiliary activation as discussed above: Auxiliary Residual Activation (ARA) and Auxiliary Sign Activation (ASA). For ARA, we replaced the first convolutional layer, which receives a residual connection from the previous block, with an ARA-Conv layer which performs weight updates using auxiliary residual activations instead of actual activations (Fig. 3a). Hence, memory overhead can be reduced for these ARA-Conv layers. The blocks in ResNet (40) can be categorized into different sets based on their dimension. For example, ResNet-50 consists of blocks with an input channel dimension of 256, 512, 1024, or 2048 as displayed in Fig. 3b. In experiments with Auxiliary Activation Learning, the blocks are replaced with ARA blocks with different strides ("sharing stride") in each set of blocks as shown in Fig. 3c and 3d. For instance, ARA(3,4,4,2) has a sharing stride of 3, 4, 4, and 2 in the sets with a dimension of 256, 512, 1024, and 2048, respectively (Fig. 3d). Therefore, the larger the sharing stride, the more blocks that are replaced by ARA blocks and the larger the amount of memory savings. Similarly, for experiments with auxiliary sign activation, the first convolutional layer was replaced by an ASA-Conv layer, which employs auxiliary sign activation

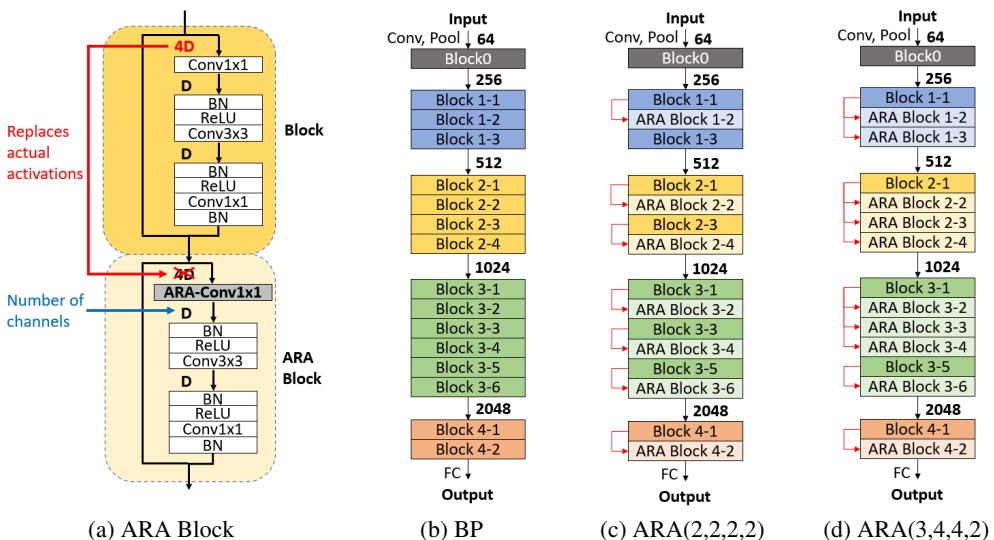

(a) ARA Block      (b) BP      (c) ARA(2,2,2,2)      (d) ARA(3,4,4,2)

Figure 3: Training ResNet-50 using Auxiliary Residual Activation (ARA).

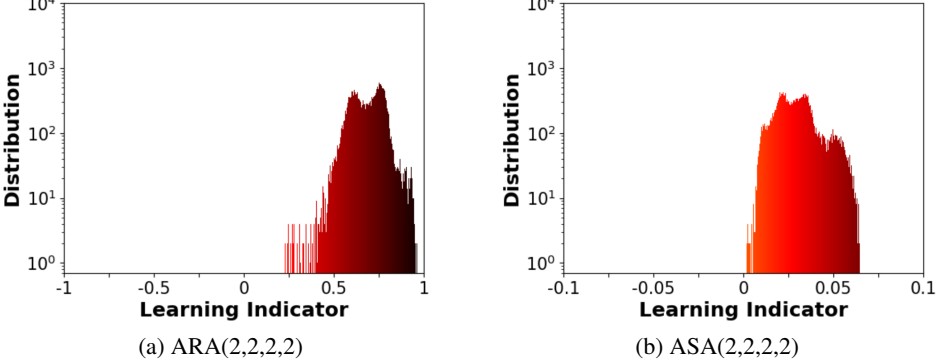

(a) ARA(2,2,2,2)          (b) ASA(2,2,2,2)

Figure 4: Distributions of learning indicator values in ResNet-18 trained on CIFAR-10.

for weight updates. Other details of the experiments including hyperparameters can be found in Appendix H.

We first trained ResNet-18 on CIFAR-10, CIFAR-100 (46), and Tiny-ImageNet (47) by using BP, ARA, and ASA to confirm the effectiveness of Auxiliary Activation Learning and the correlation between training performance and the value of the learning indicator. Fig. 4a shows that ARA not only satisfies the learning criterion (i.e., keeps learning indicator values positive), but also results in relatively large learning indicator values close to 1. This is consistent with the training performance displayed in Table 1; ARA achieves high accuracy and even outperforms backpropagation in some cases. Fig. 4b shows that ASA satisfies the learning criterion, but the values of the indicator are much lower than that of ARA. Accordingly, ASA achieves slightly lower accuracy than BP and ARA. More detailed comparisons between ARA and ASA can be found in Appendix B. In addition, the experiments in Appendix C show that the performance is significantly degraded when the learning criterion is not satisfied due to using modifications of auxiliary activation. These results suggest that the proposed learning indicator has a strong correlation with training performance.

In the next experiment, we applied ARA to training ResNet-50 and ResNet-152 on ImageNet with various sharing stride values. Experimental results are summarized in Table 2. ARA achieved competitive performance close to BP in all cases, and ARA(3,4,2,2) even achieved slightly higher performance than BP in ResNet-152. Since the mechanism of our algorithm is entirely different

Table 1: Test accuracy of ResNet-18 trained on CIFAR-10, CIFAR-100 and Tiny ImageNet.

| | Dataset | BP | ARA(2,2,2,2) | ASA(2,2,2,2) |
|---|---|---|---|---|
| Accuracy (%) | CIFAR-10 | 94.77 | 94.81 | 94.76 |
| | CIFAR-100 | 75.81 | 75.49 | 75.31 |
| | Tiny ImageNet | 58.43 | 58.46 | 57.01 |

from other memory-saving techniques, our algorithm can be employed in addition to those conventional approaches for even larger memory savings. For experiments, we trained the same networks using gradient checkpointing (GCP) (24) and ActNN (34) which is per-group activation compression training. Then, we additionally applied our ARA to these learning rules. More specifically, GCP was applied to the layers that did not employ ARA, whereas ActNN was applied to all layers including ARA_Conv layers (see Appendices H.1 and H.2 for details). Experimental results clearly show that our algorithm allows for larger memory savings with a very small increase in training time. This is in contrast with GCP and ActNN, which noticeably slows down training due to additional computations for reconstructing activations during backward propagation. For instance, applying ActNN alone to ResNet-50 reduces memory space by 11.8x while increasing training time from 18m to 49m compared to BP. Contrarily, additionally applying ARA(3,4,6,2) increases the compression rate to 15.01x without impacting training speed. We observed a similar trend in experiments under identical compression rates (see Appendix D). Furthermore, we additionally compared our ARA with Momentum ResNet (29), which is reversible network for saving training memory in Appendix E.

Table 2: Test accuracy, training memory with compression rate (bracketed), and training time (italic) of one epoch in ResNet training on ImageNet with 512 batch size and six RTX-3090 GPUs.

| Models | Baseline | - | ARA (2,2,2,2) | ARA (3,4,2,2) | ARA (3,4,4,2) | ARA (3,4,6,2) |
|---|---|---|---|---|---|---|
| ResNet-50 | BP | **76.01** 44.6 GB *17m 35s* | 75.97 (1.12x) *17m 59s* | 75.89 (1.2x) *17m 54s* | 75.62 (1.21x) *17m 55s* | 75.23 (1.22x) *18m 3s* |
| | GCP | **76.01** (2.2x) *36m 42s* | 75.97 (2.88x) *37m 2s* | 75.89 (3.44x) *37m 9s* | 75.62 (3.54x) *36m 58s* | 75.23 (3.66x) *37m 14s* |
| | ActNN | 75.96 (11.8x) *48m 48s* | 75.93 (14x) *48m 48s* | 75.67 (14.6x) *48m 10s* | 75.51 (14.8x) *47m 58s* | 75.12 (**15.01**x) *47m 08s* |
| ResNet-152 | BP | 77.38 90.2 GB *35m 37s* | 77.14 (1.16x) *36m 17s* | **77.41** (1.21x) *36m 24s* | 76.84 (1.27x) *36m 11s* | 76.64 (1.29x) *36m 30s* |
| | GCP | 77.38 (2.1x) *1h 18m* | 77.14 (2.92x) *1h 19m* | **77.41** (3.26x) *1h 19m* | 76.84 (3.75x) *1h 19m* | 76.64 (**3.95**x) *1h 20m* |

## 4.2 TRAINING TRANSFORMER, VIT, AND MLP-MIXER

Transformer (42) achieved state-of-the-art performance in NLP, and the models inspired by Transformer such as BERT (43), ViT (44), and MLP-Mixer (45) are producing promising results in computer vision. The structure of these models is quite distinct from ResNet models (40; 48). Hence, we applied our Auxiliary Activation Learning to Transformer and its variants to verify the scalability of our algorithm. We trained Transformer on IWSLT (49) and BERT-Large on MRPC (50) and MNLI (51) datasets. For ViT-Large and MLP-Mixer-Large, we used CIFAR-100 (46). Note that ARA was not used in this experiment since residual connections are connected to layer normalization layers (52) rather than to fully-connected layers, whereas ARA requires residual connections to directly connect to fully-connected layers to realize equation (13). Instead, we applied ASA to linear layers in multi-head attention and feedforward network. Fig. 5 depicts how ASA is applied to those layers. Since ASA-Linear layers store only 1-bit auxiliary sign activation instead of exact activation, memory overhead can be mitigated. We trained networks with BP and Mesa (36) which is activation compression training for Transformer-like models. Then, we additionally applied ASA to these algorithms (see Appendix H.2 for details). Mesa stores 8-bit quantized activation for linear layers;

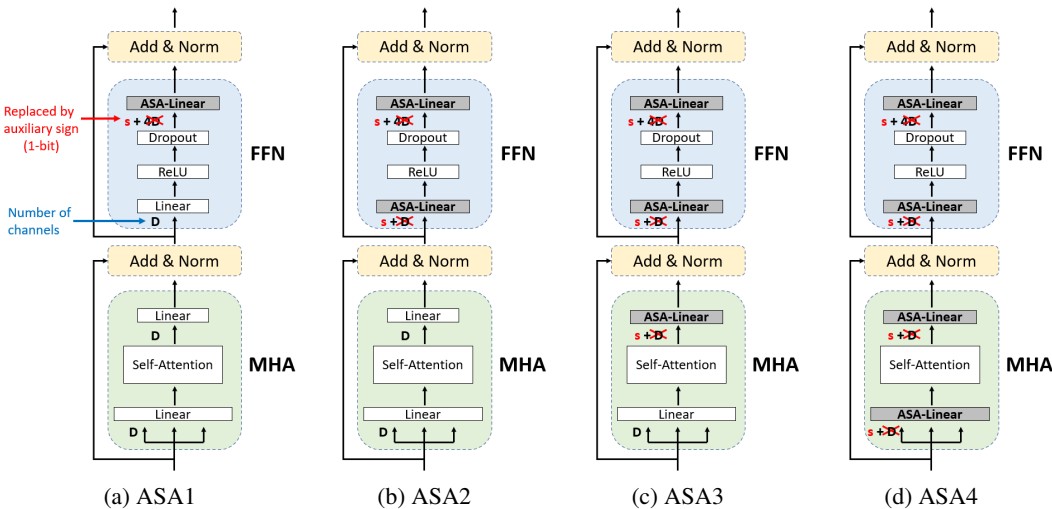

Figure 5: Transformer-like networks trained using Auxiliary Sign Activations (ASA). MHA and FFN represent multihead attention and feedforward network, respectively.

hence, applying ASA to these linear layers can further reduce memory space as ASA-Linear layers only store 1-bit sign activations. Because Mesa is designed for the networks with self-attention, it is not applied to MLP-Mixer which does not employ self-attention layers.

Experimental results are displayed in Table 3. When we apply ASA only, it closely matches or even outperforms BP while using smaller memory space. Since ASA needs to extract 1-bit sign from actual activation to construct auxiliary activation, training time is slightly increased. However, it is negligible compared to Mesa which requires quantization and dequantization during training. For example, when training BERT-Large on MNLI, BP and ASA consumed 2h 37m and 2h 41m,

Table 3: Test/validation scores, compression rate (bracketed), and training time (italic) of one epoch of Transformer, BERT, ViT, and MLP-Mixer training with 4096, 32, 512, and 256 batch sizes, respectively.

| Models | Dataset | Baseline | - | ASA1 | ASA2 | ASA3 | ASA4 |
|---|---|---|---|---|---|---|---|
| Trans-former | IWSLT | BP | 35.23
3.6 GB
*3m 15s* | 35.44
(1.1x)
*3m 21s* | 34.87
(1.1x)
*3m 23s* | 34.9
(1.2x)
*3m 26s* | 35.02
(1.3x)
*3m 38s* |
| | | Mesa | **35.45**
(1.6x)
*5m 52s* | 34.74
(1.7x)
*5m 47s* | 35.19
(1.7x)
*5m 44s* | 35.11
(1.7x)
*5m 36s* | 34.84
(**1.8**x)
*5m 8s* |
| BERT-Large | MRPC | BP | 88.56
10.9 GB
*1m 28s* | 88.69
(1.1x)
*1m 29s* | 88.23
(1.2x)
*1m 30s* | **88.97**
(1.3x)
*1m 30s* | 88.32
(1.3x)
*1m 31s* |
| | | Mesa | 88.3
(2.1x)
*2m 8s* | 88.35
(2.3x)
*2m 5s* | 88.51
(2.4x)
*2m 3s* | 88.25
(2.4x)
*2m 1s* | 88.23
(**2.5**x)
*1m 55s* |
| | MNLI | BP | 86.52
10.9 GB
*2h 37m* | **86.65**
(1.1x)
*2h 40m* | 86.49
(1.2x)
*2h 40m* | 86.42
(1.3x)
*2h 41m* | 86.39
(1.3x)
*2h 41m* |
| | | Mesa | 86.32
(2.1x)
*3h 51m* | 86.37
(2.3x)
*3h 39m* | 86.29
(2.4x)
*3h 35m* | 86.54
(2.4x)
*3h 32m* | 86.17
(**2.5**x)
*3h 24m* |
| ViT-Large | CIFAR-100 | BP | 92.93
48 GB
*2m 46s* | 92.81
(1.3x)
*2m 48s* | 92.84
(1.5x)
*2m 50s* | **92.97**
(1.5x)
*2m 51s* | 92.65
(1.6x)
*2m 52s* |
| | | Mesa | 92.89
(3.0x)
*4m 46s* | 92.75
(3.5x)
*4m 15s* | 92.72
(3.6x)
*3m 57s* | 92.95
(3.7x)
*3m 49s* | 92.84
(**4.3**x)
*3m 33s* |
| Mixer-Large | CIFAR-100 | BP | 91.62
86.8 GB
*3m 39s* | 91.45
(1.3x)
*3m 47s* | 91.72
(1.4x)
*3m 50s* | 91.66
(1.7x)
*3m 57s* | **91.91**
(**2.0**x)
*4m 1s* |

respectively, while Mesa takes 3h 51m. We can maximize memory savings by applying Mesa and our algorithm (ASA) simultaneously. If we train ViT-Large on CIFAR-100 using Mesa only, we can obtain a 3.0x compression rate. On the other hand, applying both Mesa and ASA4 simultaneously results in the compression rate of 4.3x. Furthermore, the training time is reduced by 1m 13s because linear layers are replaced with ASA-Linear layers, avoiding costly quantization and dequantization processes for these layers. Experimental results under identical compression rates can be found in Appendix D, which confirm that ASA enables faster training than Mesa with identical memory savings.

## 4.3 Usage case of Auxiliary Activation Learning

The proposed Auxiliary Activation Learning algorithm allows for training deep neural networks on the systems with limited hardware resources by alleviating memory overheads with minimal computational cost. For example, one may increase the width or depth of a neural network to improve its accuracy. In experiments, we scaled ResNet-152 and BERT-Large using the same batch size to confirm the effectiveness of our algorithm. For ResNet-152, we fixed other parameters and increased the number of layers or the width of the bottleneck block following the scheme suggested by Chen et al. (34). We also scaled BERT-Large by increasing the number of transformer blocks or the hidden size following the scheme proposed by Liu et al. (38). The results are summarized in Table 4.

Table 4: Largest models that can be trained using a single GPU with 24GB memory. (ResNet: depth = number of layers, width = width of the first bottleneck block. BERT-Large: depth = number of transformer blocks, width = hidden size.)

| Models | ResNet | | | | BERT-Large | | | |
|---|---|---|---|---|---|---|---|---|
| Learning rule | BP | ARA | ActNN | ARA+ActNN | BP | ASA4 | Mesa | Mesa+ActNN |
| Depth | 146 | 165 | 622 | 718 | 50 | 60 | 64 | 70 |
| Width | 62 | 76 | 214 | 238 | 1600 | 1728 | 1792 | 1856 |

Using ARA, we can train 13% deeper or 22% wider models compared to using BP only. Similarly, additionally applying ARA allows for using 15% deeper or 11% wider models compared to using ActNN only. For BERT-Large, we can train 20% deeper or 8% wider models by using ASA. Additionally applying ASA allows for using 9% deeper or 4% wider model compared to using Mesa only. Our algorithm may be used to increase the batch size for faster training using identical GPU resources, and experimental results are provided in Appendix F.

## 5 Discussion

In this work, we proposed a new learning rule, Auxiliary Activation Learning, which effectively alleviates memory overhead in deep neural network training. The algorithm employs an alternate form of activations for weight updates, and hence reduces the amount of data to be stored in memory without additional computations. We proved that the algorithm can train neural networks if an alternative form of activations satisfies a certain condition and also provided a performance indicator that can anticipate the training performance of the selected auxiliary activation type. Experimental results confirmed that the proposed algorithm successfully trains various deep neural networks for vision and NLP tasks with smaller memory without training speed reduction. Furthermore, since our algorithm is orthogonal to existing memory saving algorithms, it is possible to increase the compression rate further without reduction in training speed by applying existing algorithms and our algorithm simultaneously.

Since our study aims to save training memory in linear and convolutional layers, it currently cannot be applied to layers that perform other types of operations such as normalization and activation functions. In future work, we will try to address those issues to realize a more memory-efficient learning algorithm and expand its applicability.

## REPRODUCIBILITY

We presented the rationale for our algorithm in Section 3. And it was proven through the experimental results in Section 4. We have detailed various points for experimental reproduction in Section 4 and Appendix H. Furthermore, we support code to reproduce our results as follows: https://github.com/WooSunghyeon/Auxiliary_Activation_Learning.

## ACKNOWLEDGMENT

This work was supported by the National Research Foundation of Korea (Grant NRF-2022R1C1C1006880) and the Institute of Information & Communications Technology Planning & Evaluation (Grant 2021-0-01343).

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

APPENDICES

## A PROOF OF THEOREMS

**Theorem 1.** *If $r_l^T(2h_l - r_l) > 0$ in all layers, the step size $\eta$ satisfies $0 < \eta \leq \frac{1}{L}$, and the gradient of the loss function $f(W)$ is Lipschitz continuous, then $f(W)$ converges to a minimum value.*

*Proof.* Since a gradient of $f(x)$ Lipschitz continuous (i.e. $\|\nabla f(y) - \nabla f(x)\|_2 \leq L\|y - x\|_2$), it satisfies $f(y) \leq f(x) + \nabla f(x)^T(y - x) + \frac{1}{2}L\|y - x\|^2$. By substituting $W^t$ for $x$ and $W^{t+1}$ for $y$, we obtain

$$f(W^{t+1}) \leq f(W^t) + \nabla f(W^t)^T(W^{t+1} - W^t) + \frac{1}{2}L\|W^{t+1} - W^t\|^2$$

$$= f(W^t) + \nabla f(W^t)^T(-\eta\Delta W^t) + \frac{1}{2}L\|-\eta\Delta W^t\|^2$$

$$= f(W^t) - \eta[\nabla f(W^t)^T\Delta W^t - \frac{1}{2}\eta L\|\Delta W^t\|^2]$$

$$= f(W^t) - \eta K^t$$

where

$$K^t = \nabla f(W^t)^T\Delta W^t - \frac{1}{2}\eta L\|\Delta W^t\|^2$$

$$\geq \nabla f(W^t)^T\Delta W^t - \frac{1}{2}\|\Delta W^t\|^2$$

because we assumed that the step size is $0 < \eta \leq \frac{1}{L}$. By applying equations (8) and (9), we obtain

$$K^t \geq \nabla f(W^t)^T\Delta W^t - \frac{1}{2}\|\Delta W^t\|^2$$

$$= \nabla f(W^t)^T\Delta W^t - \frac{1}{2}(\Delta W^t)^T\Delta W^t$$

$$= \frac{1}{2}[2f(W^t) - (\Delta W^t)]^T\Delta W^t$$

$$= \frac{1}{2}\text{Vec}(2\delta_1 h_0^T - \delta_1 r_0^T, \ldots, 2\delta_L h_{L-1}^T - \delta_L r_{L-1}^T)^T\text{Vec}(\delta_1 r_0^T, \ldots, \delta_L r_{L-1}^T)$$

$$= \frac{1}{2}\sum_{l=0}^{L-1}\text{Vec}(\delta_{l+1}(2h_l - r_l)^T)^T\text{Vec}(\delta_{l+1}r_l^T)$$

We can show that $\text{Vec}(AB^T)^T\text{Vec}(AC^T) = \|A\|^2 C^T B$ when $A$, $B$, and $C$ are column vectors, and $B$ and $C$ have the same dimension as below.

$$\text{Vec}(AB^T)^T\text{Vec}(AC^T) = \text{Vec}([a_i b_j]_{ij})^T\text{Vec}([a_i c_j]_{ij})$$

$$= [a_1 b_1, a_2 b_1, \cdots, a_m b_n][a_1 c_1, a_2 c_1, \cdots, a_m c_n]^T$$

$$= (a_1^2 + \cdots + a_m^2)b_1 c_1 + (a_1^2 + \cdots + a_m^2)b_2 c_2 + \cdots + (a_1^2 + \cdots + a_m^2)b_n c_n$$

$$= (a_1^2 + \cdots + a_m^2)(b_1 c_1 + \cdots + b_n c_n)$$

$$= \|A\|^2 C^T B$$

Consequently, by applying this equality to the inequality above,

$$K^t = \frac{1}{2}\sum_{l=0}^{L-1}\text{Vec}(\delta_{l+1}(2h_l - r_l)^T)^T\text{Vec}(\delta_{l+1}r_l^T)$$

$$\geq \frac{1}{2}\sum_{l=0}^{L-1}\|\delta_{l+1}\|^2 r_l^T(2h_l - r_l)$$

$$\geq 0$$

because $\boldsymbol{r}_l^T(2\boldsymbol{h}_l - \boldsymbol{r}_l) > 0$ in all layers by the assumption above. Therefore, $0 \le f(\boldsymbol{W}^{t+1}) \le f(\boldsymbol{W}^t) - \eta K^t \le f(\boldsymbol{W}^{t-1}) - \eta K^{t-1} - \eta K^t \le f(\boldsymbol{W}^0) - \eta \sum_{n=0}^{t} K^n$, which suggests that the value of $f(\boldsymbol{W})$ becomes smaller as the number of epochs $t$ increases because $K^n \ge 0$. Consequently, because $f(\boldsymbol{W})$ decreases and is bounded below by 0, $f(\boldsymbol{W})$ converges to zero during training even when using alternative activations for updating weights by the monotone convergence theorem. Furthermore, if the values of $\boldsymbol{r}_l^T(2\boldsymbol{h}_l - \boldsymbol{r}_l)$ increases, $K^n$, which is the amount of loss reduction in each epoch, also increases and therefore the network converges well. □

## B COMPARISONS BETWEEN AUXILIARY RESIDUAL ACTIVATION AND AUXILIARY SIGN ACTIVATION

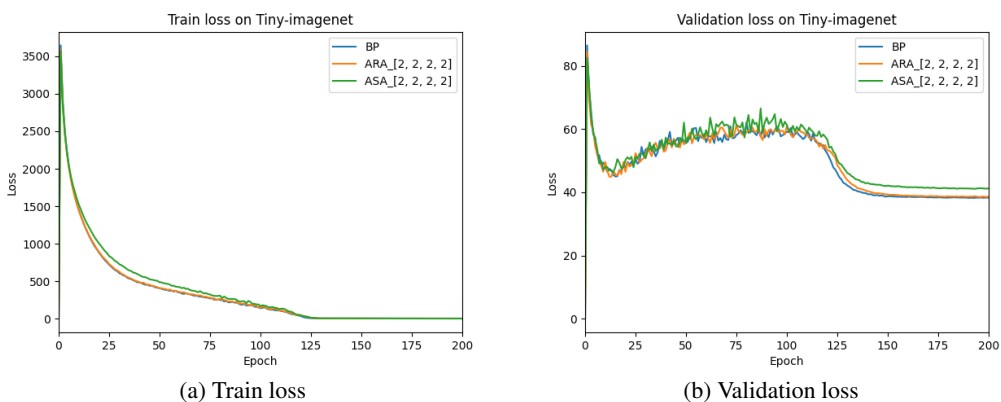

(a) Train loss

(b) Validation loss

Figure 6: Train and Validation losses in ResNet-18 training on Tiny ImageNet

Table 5 shows that the training memory of ASA is slightly larger than that of ARA since ASA has to obtain and store 1-bit auxiliary sign activation, while ARA does not need to store actual activation and just reuses auxiliary activation. We also measured the mean values of the learning indicator. The mean values of the learning indicator of ARA are significantly higher than those of ASA. Therefore, from Theorem 1, we can expect that the loss of the network would converge better by using ARA compared to ASA. This is consistent with the experimental results in Table 5; ARA achieves higher accuracy than ASA in all experiments. In addition, both train and validation losses converge better when using ARA, as shown in Fig. 6. The processing time for one epoch of ASA is slightly larger than that of ARA since ASA needs to pack 1-bit auxiliary sign activation, whereas ARA employs auxiliary residual activation as is.

Table 5: Test accuracy, training memory, training time of one epoch, and mean values of learning indicator in ResNet-18 training.

| Dataset | metric | BP | ARA (2,2,2,2) | ASA (2,2,2,2) |
|---|---|---|---|---|
| CIFAR-10 | Accuracy | 94.77 | 94.81 | 94.76 |
| | Training memory | 605 MB | 546 MB | 549 MB |
| | Training time | 20.83s | 21.30s | 21.46s |
| | Learning Indicator | - | 0.69 | 0.03 |
| CIFAR-100 | Accuracy | 75.81 | 75.49 | 75.31 |
| | Training memory | 605 MB | 546 MB | 549 MB |
| | Training time | 20.86s | 21.32s | 21.52s |
| | Learning Indicator | - | 0.71 | 0.03 |
| Tiny ImageNet | Accuracy | 58.43 | 58.46 | 57.01 |
| | Training memory | 1211 MB | 1091 MB | 1097 MB |
| | Training time | 36.82s | 37.54s | 37.94s |
| | Learning Indicator | - | 0.7 | 0.03 |

## C  DOES ADDING AUXILIARY ACTIVATION REALLY HELP?

Theorem 1 suggests that an alternative type of activation can replace actual input activation in backward propagation, as long as it satisfies the learning criterion. However, it is difficult to meet this condition using arbitrary values, so we proposed to add the alternate activation to real activation in forward propagation. Then, the input activation $h_l = o_l + a_l$ in forward propagation and the alternate activation $a_l$ in backward propagation share a common term, which is expected to help satisfy the learning criterion. In this section, we experimentally verify this claim by comparing ARA and ASA to baselines where the alternate activations are not added to real activation in forward propagation.

In baselines, we replace real activation with the previous layer's activation and 1-bit sign of original activation in backward propagation, while they are not added to real activation in forward propagation. They are denoted by Residual Activation (RA) and Sign Activation (SA), respectively. SA is identical to simply quantizing activation into 1 bit in conventional backpropagation.

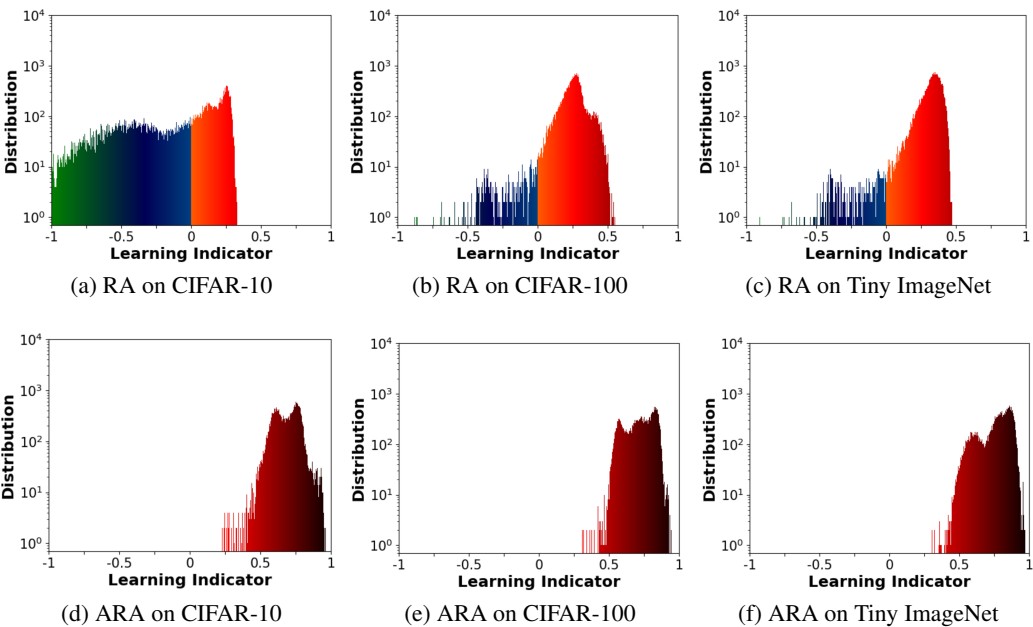

Figure 7: Distributions of learning indicator values of RA and ARA in ResNet-18

Figs. 7 and 8 display learning indicator values of ResNet-18 trained on CIFAR-10, CIFAR-100, and Tiny ImageNet. These results show that adding auxiliary activation to real activation in forward propagation significantly improves the distributions of learning indicator values by positively skewing them toward 1. This is consistent with training performance comparisons provided in Table 6. ARA and ASA noticeably outperform RA and SA in all experiments, and the performance improvement is more significant for larger datasets. Therefore, we can conclude that adding auxiliary activation to real activation in forward propagation does help neural network training.

Table 6: Test accuracy of ResNet-18 trained on CIFAR-10, CIFAR-100, and Tiny ImageNet.

|  | Learning Rule | CIFAR-10 | CIFAR-100 | Tiny ImageNet |
|---|---|---|---|---|
|  | BP | 94.77 | 75.81 | 58.43 |
|  | RA (2, 2, 2, 2) | 94.1 | 72.23 | 53.91 |
| Accuracy (%) | ARA (2, 2, 2, 2) | 94.81 | 75.49 | 58.46 |
|  | SA (2, 2, 2, 2) | 94.36 | 74.58 | 55.76 |
|  | ASA (2, 2, 2, 2) | 94.76 | 75.31 | 57.01 |

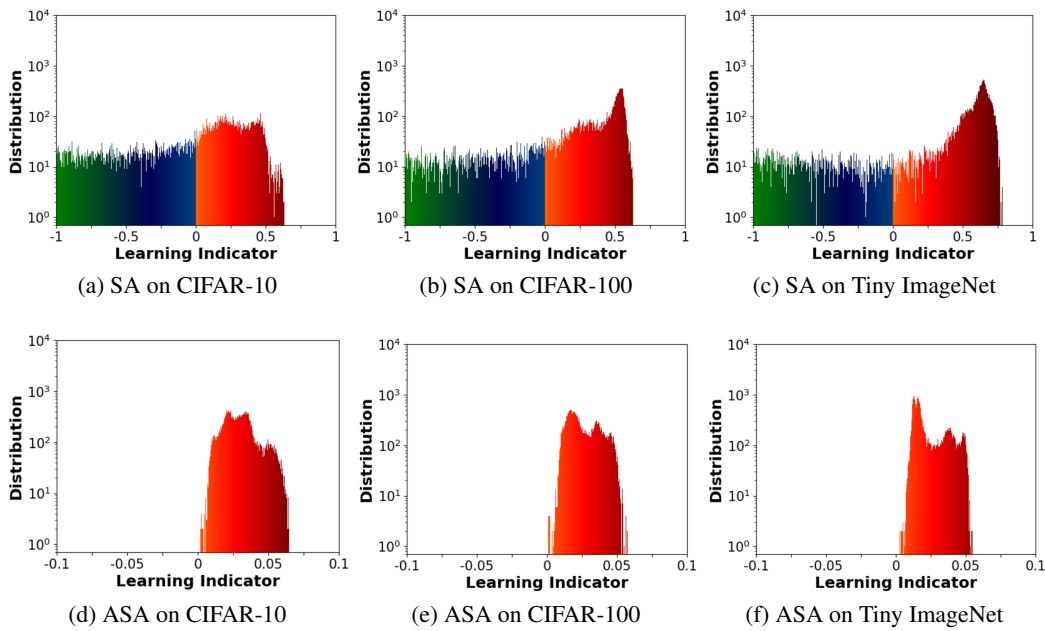

Figure 8: Distributions of learning indicator values of SA and ASA in ResNet-18

# D   COMPARISONS BETWEEN AUXILIARY ACTIVATION LEARNING AND OTHER MEMORY SAVING ALGORITHMS UNDER IDENTICAL COMPRESSION RATES

In this section, we compare our Auxiliary Activation Learning with other memory saving algorithms under identical compression rates. We apply GCP to the last Conv-BN layer of residual blocks in Fig. 3a because the dimension of input activation in the last batch normalization layer is equal to that of the ARA-Conv layer, which is not stored in memory during ARA. The input activation of the batch normalization layer is not stored in Conv-BN layers, but it is recomputed from the input activation of the convolutional layer during backward propagation. Through this, an identical amount of memory saving can be achieved by using GCP. For ActNN, we applied it to first convolutional layer of residual blocks as we did for ARA in Fig. 3a, which resulted in an identical compression rate. While our ASA stores 1-bit auxiliary sign activation, Mesa stores 8-bit compressed activation. Therefore, we cannot achieve the same memory saving by applying Mesa in the same way as ASA. To overcome this issue, we additionally apply Mesa to the batch normalization and layer normalization layers to obtain the same compression rate to ASA. Tables 7 and 8 show that ARA and ASA train the networks faster than other memory-saving algorithms with identical compression rates as they do not require additional costly computations.

Table 7: Test accuracy, training memory, compression rate (bracketed), and training time of one epoch in ResNet-152 training on ImageNet with 512 batch sizes.

|  | BP | ARA (3,4,2,2) | GCP | ActNN |
|---|---|---|---|---|
| Accuracy | 77.38 | 77.41 | 77.38 | 77.35 |
| Training memory | 90.2 GB | 74.5 GB (x1.21) | 74.4 GB (x1.21) | 74.5 GB (x1.21) |
| Training time | 35m 37s | 36m 24s | 45m 21s | 47m 39s |

Table 8: Test accuracy, training memory, and training time of one epoch in ViT-Large training on CIFAR-100.

|  | BP | ASA3 | Mesa |
|---|---|---|---|
| Accuracy | 92.93 | 92.97 | 92.91 |
| Training memory | 48 GB | 33.1 GB (x1.5) | 31.8 GB (x1.5) |
| Training time | 2m 46s | 2m 51s | 3m 12s |

# E    COMPARISONS BETWEEN AUXILIARY RESIDUAL ACTIVATION AND MOMENTUM RESNET.

In this section, we compare ARA with Momentum ResNet (MResNet) (29), which is a reversible network that can be applied to diverse networks with residual connections. For comparisons, we trained ResNet-50 on CIFAR-100 with a 5e-2 learning rate for 100 epochs with 10 warm-up steps and cosine annealing. The results are displayed in Table 9. First, ARA closely matches BP in accuracy, but MResNet exhibits a small accuracy degradation. While ARA requires larger training memory than MResNet, ARA is significantly faster since MResNet needs to recalculate the input of residual blocks along with additional forward propagation during backward propagation.

Table 9: Test Accuracy, training memory, and training time of one epoch in ResNet-50 with 128 batch sizes.

| Dataset | BP | MResNet | ARA (3,4,2,2) |
|---|---|---|---|
| Accuracy | 76.83 | 76.27 | 76.78 |
| Training memory | 3.30 GB | 2.54 GB | 2.71 GB |
| Training time | 51.54s | 2m 21s | 53.49s |

# F    MAXIMUM BATCH SIZE AND TRAINING TIME WITH AUXILIARY ACTIVATION LEARNING.

We trained various networks using only three RTX-3090 GPUs to confirm the effectiveness of our algorithm. The experimental results for ResNet-152 and ViT-Large are displayed in Tables 10 and 11. Using memory-saving techniques allows for a larger batch size for a given GPU memory space. Interestingly, although conventional algorithms such as GCP, ActNN, and Mesa enables a larger batch size, it does not translate to faster training speed due to additional computations. In contrast, our ARA and ASA algorithms take full advantage of larger batch size and improve training speed.

Table 10: Maximum batch size and training time of ResNet-152 on ImageNet. The sharing stride of ARA is (3,4,6,2).

|  | BP | GCP | ActNN | ARA | GCP + ARA | ActNN + ARA |
|---|---|---|---|---|---|---|
| Max batch size | 350 | 700 | 1950 | 450 | 1250 | **2200** |
| Training time | 49m | 1h 4m | 56m | **47m** | 57m | 52m |

Table 11: Maximum batch size and training time of ViT-Large on CIFAR-100.

|  | BP | Mesa | ASA4 | Mesa + ASA4 |
|---|---|---|---|---|
| Max batch size | 600 | 1600 | 900 | **2100** |
| Training time | 1m 53s | 2m 9s | **1m 49s** | 1m 59s |

# G    COMPARISONS BETWEEN AUXILIARY ACTIVATION LEARNING AND SM3

In this section, we compare ARA with SM3 (14), which is a memory-efficient optimizer. While our algorithm aims to reduce the amount of input activation to be stored, SM3 aims to reduce the memory required for storing states generated by accumulating previous gradients of parameters.

For fair comparisons, we measured the total memory allocated to a GPU instead of the memory required to store activations. Experimental results are summarized in Table 12. The results show that our ASA4 algorithm requires smaller memory space but takes more time to process one epoch compared to SM3. This is because the SM3 optimizer is faster than the Adam optimizer that we used for all experiments. Since the theoretical backgrounds behind these two methods are quite different, they can be combined together to obtain even larger savings. As displayed in Table 12, if we apply both methods simultaneously, we can obtain the largest memory savings with minimal impact on the accuracy.

Table 12: Test Accuracy, training memory, and training time of one epoch in Bert-Large on MRPC with 32 batch sizes.

| Dataset | BP | SM3 | ASA4 | SM3 + ARA |
|---------|------|------|------|-----------|
| Accuracy | 88.56 | 88.23 | 88.32 | 88.15 |
| Total memory | 22.2 GB | 19.9 GB | 19.1 GB | 16.55GB |
| Training time | 1m 28s | 1m 20s | 1m 31s | 1m 22s |

## H  EXPERIMENTAL DETAILS

### H.1  HOW WE APPLY GRADIENT CHECKPOINITING IN EACH EXPERIMENT

Chen et al.(24) suggested that when there is a BN-ReLU-Conv layer in the network, applying gradient checkpointing to those layers can reduce memory requirements without incurring significant computational overhead. More specifically, only the input activation of the BN layer is stored during forward propagation, and the rest is recalculated in backward propagation. Therefore, while the BN and ReLU layers have to be recomputed during backward propagation to generate the input activation of the Conv layer, there is no need to recompute the Conv layer itself which may lead to large computational overhead. We applied this method to BN-ReLU-Conv3x3 layers in Fig. 3a to reduce memory overhead without a significant amount of recomputation. In addition, we apply gradient checkpointing to BN-ReLU-Conv1x1-BN layers in Fig. 3a. Although we have to recompute Conv1x1, its computational overhead is relatively small compared to Conv3x3 layers. We also applied gradient checkpointing to the first Conv-BN-ReLU-MaxPool layers in ResNet. Here only the input of the first Conv layer is stored during forward propagation, and the input activation of the other layers is recomputed in backward propagation. Although the first Conv layer has to be recomputed, a large amount of training memory can be saved because the amount of the input activation in the first Conv, BN, and MaxPool layers is large.

### H.2  HOW WE APPLY ACTIVATION COMPRESSION TRAINING IN EACH EXPERIMENT

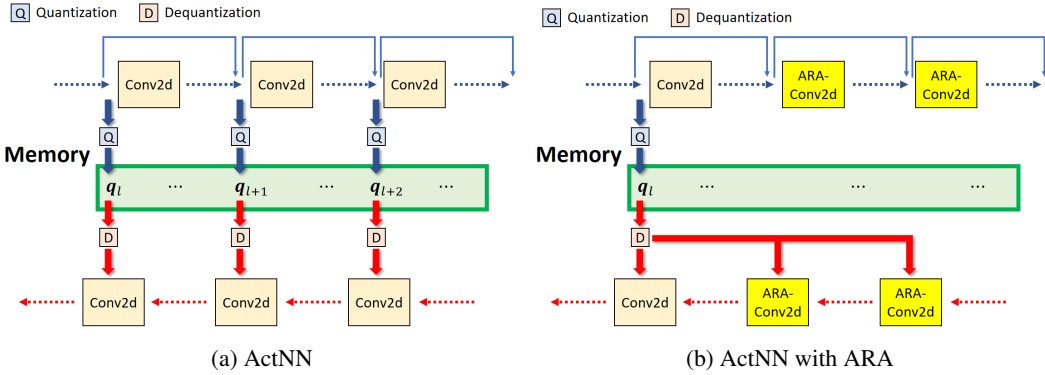

(a) ActNN                                        (b) ActNN with ARA

Figure 9: Training process of ActNN with ARA

ActNN (34) stores 2-bit compressed activations instead of actual activations for all layers except for ReLU and pooling which only needs 1-bit sign activations. We applied ActNN (L3) to all layers,

which performs per-group quantization with fine-grained mixed precision. The per-group quantization partitions the activation into several groups, and then quantization is performed by groups. The fine-grained mixed precision includes two processes: per-sample allocation and per-layer allocation. In per-sample allocation, the sensitivity, which is related to gradient variance, is calculated for each group and then an optimal number of bits for each group is determined using this sensitivity. Then, per-layer allocation calculates the sensitivity of each layer and then an optimal number of bits for each layer is obtained. When ARA and ActNN are applied simultaneously, we perform 2-bit group-wise quantization and per-sample allocation on auxiliary residual activation. However, per-layer allocation is not applied to auxiliary residual activation because it is not straightforward to calculate the sensitivity of ARA layers. While ActNN needs to quantize and store activations of all layers, we can skip this process for ARA_Conv layers when using ActNN with ARA (Fig. 9).

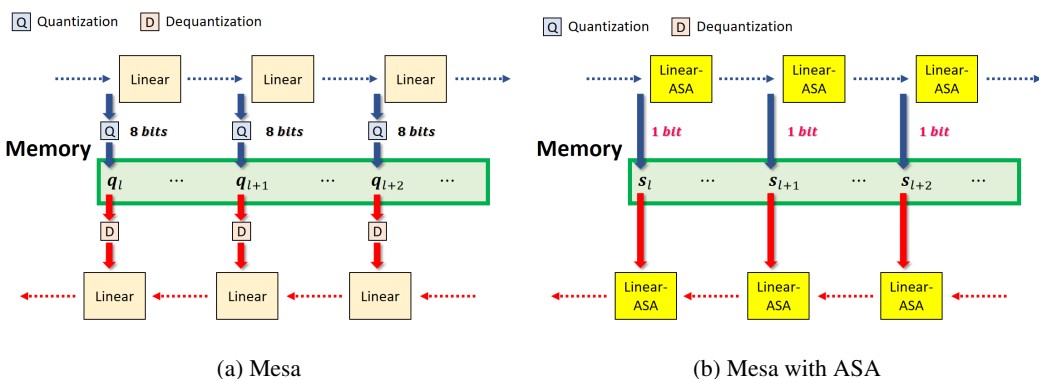

(a) Mesa            (b) Mesa with ASA

Figure 10: Training process of Mesa with ASA.

Pan et al. (36) pointed out that conventional activation compression algorithms do not consider heterogeneous activation distributions in a multi-head attention layer and therefore they are not applicable to Transformer-like models. Therefore, they proposed Mesa which performs head-wise quantization instead of per-tensor quantization. When we combine Mesa and ASA, we replace linear layers with ASA-Linear layers as shown in Fig. 10.

## H.3 DATASETS

**CIFAR-10 and CIFAR-100**   The CIFAR-10 dataset (46) consists of 50,000 training images and 10,000 test images which are 32x32 RGB images for 10-class image classification. Likewise, the CIFAR-100 dataset (46) includes 50,000 training images and 10,000 test images with the same resolution for 100-class image classification.

**Tiny ImageNet**   The Tiny-ImageNet dataset (47) consists of images of 200 classes and each class has 500 images for training. It also contains 10,000 test images. All images included in Tiny ImageNet are selected from ImageNet and downsized to 64x64.

**ImageNet**   The ImageNet dataset (41) is a representative large-scale dataset for image classification. It consists of 1,281,167 training images, 50,000 validation images, and 100,000 test images which are all RGB images. The average image size on ImageNet is 469x387, but they are randomly cropped to 224x224 for image classification.

**IWSLT 2016**   The IWSLT 2016 dataset (49) is constructed for spoken language translation task. We choose IWSLT2016 De-En dataset which is used for German-to-English translation. It has 1,611 talks, 197K sentences, and 3.96M tokens as a training set while the validation set consists of 13 talks, 1.1K sentences, and 21K tokens.

**MRPC**   The Microsoft Research Paraphrase Corpus (MRPC) dataset (50) consists of 5081 sentence pairs from newswire articles. In this dataset, the training set contains 4076 sentence pairs with 2753 paraphrases and the test set contains 1725 pairs with 1147 paraphrases.

**MNLI** The Multi-Genre Natural Language Inference (MultiNLI) dataset (51) consists of 433k sentence pairs for ten genres (Face-to-Face, Telephone, etc.). The dataset is divided into 392702 train sets, 9815 validation_matched sets which are subsets of train sets, and 9832 validation_mismatched sets which are included in train sets.

## H.4 HYPERPARAMETES

**ResNet** For training ResNet-18 from scratch on CIFAR-10, CIFAR-100, and Tiny ImageNet, we set the batch size and the total number of epochs to 128 and 200, respectively. We applied stochastic gradient descent with momentum (53) along with weight decaying (46) to our experiments. The momentum and weight decay rate was set to 0.9 and 1e-4, respectively. The learning rate of the layers except for ASA layers was scheduled by cosine annealing (54) with a 0.1 initial learning rate during 200 epochs. In comparison, we used a 100x higher learning rate for ASA layers to make the magnitude of weight updates comparable to those of other layers. We also set $\epsilon$ in equations (15) and (16) to 0.01. We used a nearly identical set of hyperparameters for ImageNet experiments. However, we set the batch size and the initial learning rate to 512 and 5e-2, respectively, and the number of epochs is 90 with 4 warm-up steps.

**Transformer** In Transformer, the batch size and the number of epochs are set to 4096 and 20, respectively. We trained Transformer from scratch on IWSLT and used the Adam optimizer (55) by setting $\beta_1$ to 0.9, $\beta_2$ to 0.98, and $\epsilon$ to 1e-9. In addition, the learning rate was updated by the warm-up learning rate scheduler (54), and the number of warm-up steps was set to 4,000. We applied a 100x higher learning rate to ASA layers as in ResNet experiments. $\epsilon$ in equations (15) and (16) for ASA was set to 0.01. The model dimension was 512, the hidden layer dimension was 2048, the number of heads was 8, and the dropout rate (56) was 0.2. Furthermore, we applied label smoothing with a 0.1 label smoothing rate (57) to all experiments in Transformer.

**BERT-Large** We used Bert-Large (uncased) pretrained on BookCorpus (58) and English Wikipedia for fine-tuning on MRPC and MNLI. The batch size was set to 32, and we trained BERT-Large for three epochs. The Adam optimizer (55) was used with 0.9 $\beta_1$, 0.999 $\beta_2$, 1e-6 $\epsilon$, and 0.01 weight decay rate. We also employed warm-up in the learning rate up to 10% of the total step, and cosine annealing is used for the remaining epochs (54). The learning rate of cosine annealing was set to 1.5e-5. These learning rate was increased by 100x for ASA layers in ViT experiments. $\epsilon$ in equations (15) and (16) was set to 0.01. The other hyperparameters were selected according to the original BERT paper (43).

**ViT-Large** We fine-tuned a ViT-Large model pretrained on ImageNet-21k. The model was fine-tuned using 32 batch size on CIFAR-100. We set the training batch size to 512 and experimented with different learning rates of 3e-2, 1e-1, and 3e-1 to achieve the highest performance. For training, learning rate was adjusted by stochastic gradient descent with momentum (53) and a warm-up cosine scheduler (54) was applied. The model was trained for 1000 steps with 100 warm-up steps. For Linear-ASA layers, 100x larger learning rate was used. We also set $\epsilon$ in equations (15) and (16) to 0.01. The remaining hyperparameters were set according to the original ViT paper (44).

**MLP-Mixer-Large** The MLP-Mixer-Large model was fine-tuned from a pretrained model on ImageNet-21k with 16 patch size. We trained the model on CIFAR-10 for 2000 steps with 56 batch size. Like ViT, the SGD optimizer (53) was applied to these experiments. We set the weight decay rate and momentum to 0 and 0.9, respectively. During the first 200 steps, the learning rate warmed up and cosine annealing (54) was employed in the remaining epochs. We set the learning rate to 3e-2, 1e-1, and 3e-1. In the ASA layers, however, the learning rate was increased by 100x like Transformer and ViT-Large. $\epsilon$ of ASA layers in equations (15) and (16) was set to 0.01.The other hyperparameters followed the original MLP-Mixer paper (45).

## H.5 DEVICES AND MEMORY MEASUREMENTS

In all experiments, we used Nvidia GeForce RTX 3090 GPUs. While one GPU was utilized to perform most of the experiments, we used six GPUs to train ResNet on ImageNet to reduce training time. We calculated the amount of training memory using *torch.cuda.memory_allocated* function

in PyTorch instead of *nvidia-smi* because *nvidia-smi* includes unused memory space handled by the memory allocator in the report[1].

# I  MEMORY AND COMPUTE COMPLEXITY OF AUXILIARY ACTIVATION LEARNING

Table 13: The compute and memory complexity analysis.

|                     | BP    | GCP        | AAL only | AAL with GCP   |
|---------------------|-------|------------|----------|----------------|
| Compute complexity  | $3N$  | $4N$       | $3N$     | $4N$           |
| Memory complexity   | $N$   | $2\sqrt{N}$ | $N - A$  | $2\sqrt{N - A}$ |

In this section, we compare the compute and memory complexity of different learning algorithms. For the sake of simplicity, we assume that the computational cost of forward propagation and backward propagation of a single layer is 1. The amount of activations of a single layer is also assumed to be 1. Then we calculate the computational cost and memory space required for training a neural network with $N$ identical layers. BP would need $N$ computations for forward propagation, $N$ computations for error propagation, and $N$ computations for calculating weight updates, which results in $3N$ computations in total. For memory space, BP has to store $N$ activations generated during forward propagation. On the other hand, GCP requires $N$ additional computations for the recomputation of activations. However, it can reduce the memory complexity to $2\sqrt{N}$ when the network is divided into $\sqrt{N}$ blocks as suggested in (24). In our Auxiliary Activation Learning (AAL), we have to store auxiliary activations instead of actual activations. We assume that our algorithm is applied to $A$ layers in the network. Then, we have to store actual activation for the rest of the layers, which translates to a memory complexity of $N - A$. For the layers employing our algorithm, we only have to store auxiliary activations and its amount is negligible compared to real activations. For example, Auxiliary Residual Activation (ARA) uses the previous layer's activation as auxiliary activation, and hence there is no need to store this auxiliary activation since it is already stored in memory during forward propagation of the previous layer. Auxiliary Sign Activation (ASA) has to store a 1-bit sign of actual activations, respectively, but they are also significantly smaller than real activations. Therefore, our algorithm has $N - A$ memory complexity and $3N$ compute complexity. To further reduce memory complexity, we can apply gradient checkpointing to the layers where we do not apply our AAL algorithm. Thus, the memory complexity of those layers would decrease to $2\sqrt{N - A}$ and the compute complexity becomes $4N$ due to recomputation.

# J  COMPARISONS TO ALGORITHMS TO APPROXIMATE BACKPROPAGATION

Previous studies have suggested various algorithms to approximate backpropagation (59; 60; 61). Our algorithm may be considered in line with such approaches as it replaces actual input activation with auxiliary activation. Furthermore, Uniform Sign-Concordant Feedback (61), which uses the sign of forward weights as feedback weights, looks similar to the concept of ASA which uses the sign of activation as auxiliary activation. However, the motivation is quite different. In prior works on approximate backpropagation, the authors aim to remove the need for symmetric feedforward and feedback weights ("weight transport problem") by approximating the feedback path for better bio-plausibility. Contrarily, our algorithm reduces memory requirements by using an alternate form of activations, but it does not necessarily improve bio-plausibility as we still need symmetric forward and backward paths.

---

[1]https://pytorch.org/docs/stable/notes/cuda.html?highlight=buffer#memory-management

# K CONVERGENCE SPEED OF AUXILIARY ACTIVATION LEARNING

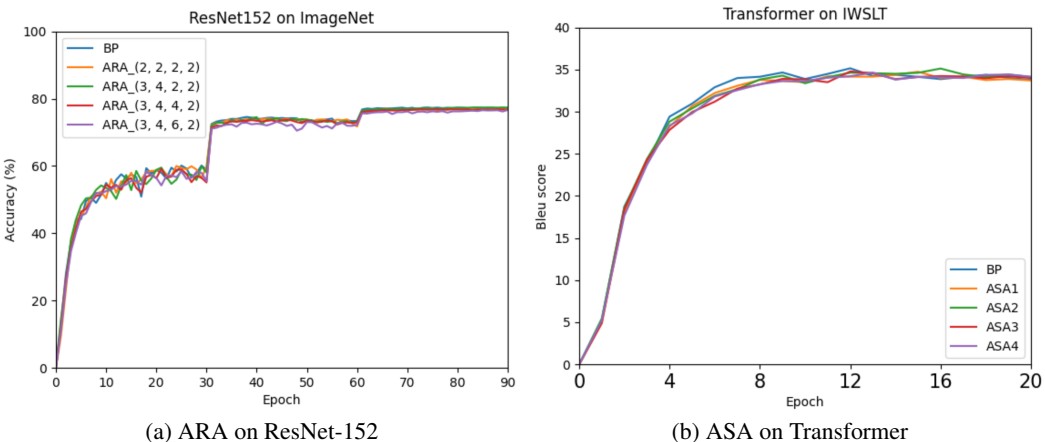

(a) ARA on ResNet-152          (b) ASA on Transformer

Figure 11: Learning curves of BP and Auxiliary Activation Learning.

