# OpenReview forum: "Learning with Auxiliary Activation for Memory-Efficient Training"
_ICLR.cc/2023/Conference — ICLR 2023 poster_

### Official Review · Reviewer_YGpd · 2022-10-24

**Confidence:** 4
**Correctness:** 2
**Technical Novelty And Significance:** 3
**Empirical Novelty And Significance:** 2
**Recommendation:** 6

**Clarity, Quality, Novelty And Reproducibility:**

The paper is clearly written and there is a link to code in Github.
Quality got me puzzled when considering the cost function as convex. Is Bishop wrong?

**Strength And Weaknesses:**

Strengths:
- The proposed strategy seems to reduce memory space (11.8x) for ResNet, while not having a big influence in training time.
- Extensive evaluation on training times and memory savings for Vision (ResNet) and NLP architectures (BERT, Transformers).

Weaknesses:
- “For mathematical analysis, we consider a loss function f(x) which is CONVEX, differentiable, and Lipschitz continuous.” Good luck with that. This is a very strong constraint and an unfeasible one when talking about Neural Networks.
- It only compares with GCP (Chen et al., 2016) and ActNN (Chen et al. 2021), but it failed to compare to recent state-of-the-art methods, most notably Momentum ResNets (Sander et al., 2021). And looking at the Momentum ResNets paper it would seem to me that Momentum ResNets provide much less memory requirements and they provide higher accuracy than what is reported in table 1.
- Memory savings are usually reported in parameter requirements, here it just says "reduce memory space (11.8x) for ResNet" and tables 2 and 3 show minimal memory reduction when compared to backpropagation (1.1x to 1.3x), such memory space is when compared to ActNN (much higher overhead).

**Summary Of The Paper:**

This paper proposes a new learning rule, named Auxiliary Activation Learning to reduce memory requirements of deep neural networks

**Summary Of The Review:**

The paper seems to be well written an experimentation is exhaustive. The mathematical proof seems to be flawed as the convexity constraint would practically not apply to any deep neural network. Or maybe I misunderstood the message authors try to convey here.
Experimental evaluation even is extensive misses the comparison with recent state-of-the-art work which seem to be better at reducing memory requirements and are better in accuracy.

---

> ### Author Response · Authors · 2022-11-13
> **Response to Reviewer YGpd**
>
> [1] Chen, J., Zheng, L., Yao, Z., Wang, D., Stoica, I., Mahoney, M., & Gonzalez, J. (2021, July). Actnn: Reducing training memory footprint via 2-bit activation compressed training. In International Conference on Machine Learning (pp. 1803-1813). PMLR.
>
> [2] Evans, R. D., & Aamodt, T. (2021). AC-GC: Lossy Activation Compression with Guaranteed Convergence. Advances in Neural Information Processing Systems, 34, 27434-27448.
>
> [3] Liu, X., Zheng, L., Wang, D., Cen, Y., Chen, W., Han, X., ... & Cheung, A. (2022, June). GACT: Activation compressed training for generic network architectures. In International Conference on Machine Learning (pp. 14139-14152). PMLR.
>
> [4] Huo, Z., Gu, B., & Huang, H. (2018, July). Decoupled parallel backpropagation with convergence guarantee. In International Conference on Machine Learning (pp. 2098-2106). PMLR.
>
> [5] Huo, Z., Gu, B., & Huang, H. (2018). Training neural networks using features replay. Advances in Neural Information Processing Systems, 31.
>
> [6] Belilovsky, E., Eickenberg, M., & Oyallon, E. (2020, November). Decoupled greedy learning of cnns. In International Conference on Machine Learning (pp. 736-745). PMLR.
>
> [7] Sander, M. E., Ablin, P., Blondel, M., & Peyré, G. (2021, July). Momentum residual neural networks. In International Conference on Machine Learning (pp. 9276-9287). PMLR.
>
> [8] Chen, T., Xu, B., Zhang, C., & Guestrin, C. (2016). Training deep nets with sublinear memory cost. arXiv preprint arXiv:1604.06174.
>
> [9] Chakrabarti, A., & Moseley, B. (2019). Backprop with approximate activations for memory-efficient network training. Advances in Neural Information Processing Systems, 32.
>
> [10] Pan, Z., Chen, P., He, H., Liu, J., Cai, J., & Zhuang, B. (2021). Mesa: A Memory-saving Training Framework for Transformers. arXiv preprint arXiv:2111.11124.
>
> [11] Gomez, A. N., Ren, M., Urtasun, R., & Grosse, R. B. (2017). The reversible residual network: Backpropagation without storing activations. Advances in neural information processing systems, 30.

---

> ### Author Response · Authors · 2022-11-13
> **Response to Reviewer YGpd**
>
> **Q4) tables 2 and 3 show minimal memory reduction when compared to backpropagation (1.1x to 1.3x), such memory space is when compared to ActNN (much higher overhead).**
>
> **A4)** We thank the reviewer for pointing out a crucial point. We agree with the reviewer that our algorithm may achieve lower memory savings than some prior methods. However, we would like to emphasize that our work is meaningful in that it suggests an entirely new way to approximate activation. More specifically, prior approaches reduce training memory through gradient checkpointing [8], quantizing activation [1-3, 9, 10], or using reversible neural network architectures [7, 11], whereas our algorithm replaces actual activation with auxiliary activation. As a result, the proposed algorithm exhibits three advantages: i) minimal impact on the training speed, ii) high degree of applicability, and iii) being orthogonal to other techniques.
>
> First, prior methods degrade the training speed because they require additional computation. For instance, GCP and reversible networks require recomputing activations during backward propagation. ActNN and Mesa need to process additional operations such as computing min/max values, stochastic rounding, tensor reshaping, and Hadamard product for quantization/dequantization. In addition, ActNN additionally needs a greedy algorithm to allocate bits for mixed-precision quantization. In our experiments, we used the authors' open-source implementation based on CUDA to maximize speed for ActNN [1] and Mesa [10], a large training speed reduction was still observed, as shown in Tables R4.2 and R4.3. This trend is also consistent with the observation provided in previous studies; for instance, the authors in [1, 3, 10] indicate that the training speed could be significantly reduced when using their methods. Contrarily, our ARA does not require this additional operation for training, resulting in substantially faster training than the other algorithms. Although ASA needs to pack auxiliary sign activation, it is implemented using a simple bitwise operation, resulting in a substantially smaller impact on the training speed.
>
> Second, some prior studies report significant memory savings by modifying neural network architectures into reversible structures. However, this is not applicable to other types of neural networks since it is not always straightforward to transform a well-performing non-reversible model into a reversible one. On the other hand, our algorithm is applicable to a wide range of neural network models. ARA can be used on any model with residual connections, and ASA can be applied to all networks without restriction.
>
> Finally, the theoretical background of our algorithm is different from other memory-saving techniques. As a result, our algorithm can be combined with other methods for even larger memory savings. For instance, our ARA and ASA could be successfully combined with GCP, ActNN, and Mesa to reduce training memory further, as demonstrated in the experimental results in the manuscript.
>
> **Table R4.2. Test accuracy, training memory, and training time of one epoch in ResNet-152 training on ImageNet with 512 batch sizes.**
>
> | | BP | ARA (3,4,2,2) | GCP | ActNN |
> | --- | --- | --- | --- | --- |
> | Accuracy | 77.38 | 77.41 | 77.38 | 77.15 |
> | Training memory | 90.2 GB | 74.5 GB (x1.21) | 74.4 GB (x1.21) | 74.5 GB (x1.21) |
> | Training time | 35m 37s | 36m 24s | 45m 21s | 47m 39s |
>
> **Table R4.3. Test accuracy, training memory, and training time of one epoch in ViT-Large training on CIFAR-100 with 128 batch sizes.**
>
> | | BP | ASA3 | Mesa |
> | --- | --- | --- | --- |
> | Accuracy | 92.93 | 92.97 | 92.91 |
> | Training memory | 48 GB | 33.1 GB (x1.5) | 31.8 GB (x1.5) |
> | Training time | 2m 46s | 2m 51s | 3m 12s |

---

> ### Author Response · Authors · 2022-11-13
> **Response to Reviewer YGpd**
>
> **Q3) Memory savings are usually reported in parameter requirements, here it just says "reduce memory space (11.8x) for ResNet".**
>
> **A3)** We thank the reviewer for valuable feedback. In the initial submission, we reported the compression rate to be consistent with prior work on other memory-saving techniques [1-3] that also employed the same metric for comparisons. We suspect that the reviewer is suggesting to include the absolute amount of memory space. We have added those numbers to all tables in the revised manuscript. If the reviewer would like to see other types of statistics, we will also include those data through revision.

---

> ### Author Response · Authors · 2022-11-13
> **Response to Reviewer YGpd**
>
> **Q2) It only compares with GCP (Chen et al., 2016) and ActNN (Chen et al. 2021), but it failed to compare to recent state-of-the-art methods, most notably Momentum ResNets (Sander et al., 2021). And looking at the Momentum ResNets paper it would seem to me that Momentum ResNets provide much less memory requirements and they provide higher accuracy than what is reported in table 1.**
>
> **A2)** We thank the reviewer for bringing related work to our attention. Table 2 in the Momentum ResNets paper [7] reports that the test accuracy of BP and MResNet is 76.86% and 76.39%, respectively, when training MresNet-101 on CIFAR-100. On the other hand, Table 1 in our paper reports experimental results for training a much smaller model, ResNet-18, on CIFAR-100, and the test accuracy of BP and MResNet is 75.81% and 75.49%. Also, the hyperparameters (e.g., the number of training epochs and the learning rate) are different, making comparisons cumbersome. Therefore, for fair comparisons, we applied Momentum ResNet and our algorithm to ResNet-50 training on CIFAR-100. Experimental results are summarized in the table below.
>
> **Table R4.1. Test accuracy, training memory, and training time of one epoch in ResNet-50 training on CIFAR-100 with 128 batch sizes.**
>
> | | BP | MResNet | ARA (3,4,2,2) |
> | --- | --- | --- | --- |
> | Accuracy | 76.83 | 76.27 | 76.78 |
> | Training memory | 3.30 GB | 2.54 GB | 2.71 GB |
> | Training time | 51.54s | 2m 21s | 53.49s |
>
> First, ARA closely matches BP in accuracy, but MResNet exhibits a small accuracy degradation. This is consistent with the results reported in Table 2 in [7]. While ARA requires larger training memory than MResNet, ARA is significantly faster since MResNet needs to recalculate the input of residual blocks along with additional forward propagation during backward propagation. These points have been added as a new section (Appendix E) in the revised manuscript.

---

> ### Author Response · Authors · 2022-11-13
> **Response to Reviewer YGpd**
>
> We thank the reviewer for carefully reviewing our submission and providing valuable comments. Please find our responses below:
>
> **Q1) "For mathematical analysis, we consider a loss function f(x) which is CONVEX, differentiable, and Lipschitz continuous." Good luck with that. This is a very strong constraint and an unfeasible one when talking about Neural Networks.**
>
> **A1)** We thank the reviewer for pointing out an important point. First of all, we apologize that there was a mistake in the initial draft. We initially assumed that
>
> $f$ **is convex, differentiable and Lipschitz continuous**
>
> , so that Theorem 1 can be proven using the quadratic upper bound as shown in Appendix A.
>
> **(quadratic upper bound)** $f(y) \leq f(x) + \nabla f(x)^{T}(y-x) + \frac{L}{2}\lVert y-x \rVert_{2}^{2}$
>
> However, the inequality above actually holds for non-convex functions as well. Therefore, in the revised manuscript, we have corrected our assumption as follows:
>
> **The gradient of $f(x)$ is Lipschitz continuous**
>
> Under the assumption above, we can induce the quadratic upper bound even if is a non-convex function. Detailed proof can be found at the link below:
>
> (proof) https://drive.google.com/file/d/10L0DDOrD0IBoP8Zd7ZiaHZRqpBmbqmnc/view?usp=share_link
>
> Please note that the constraint of the gradient of being Lipschitz continuous has been widely used in prior work on analyzing deep neural network training and reducing training memory [1-6]. Therefore, we believe that we are taking an identical approach to prior work to prove the effectiveness of the proposed algorithm; we first theoretically show that it works under some constraints, which were also employed in prior work, and experimentally demonstrate its performance using various models.

---

### Official Review · Reviewer_zs6z · 2022-10-24

**Confidence:** 3
**Clarity, Quality, Novelty And Reproducibility:** NA
**Correctness:** 3
**Technical Novelty And Significance:** 3
**Empirical Novelty And Significance:** 3
**Recommendation:** 6

**Strength And Weaknesses:**

Strength:
The paper proposes a novel algorithm to achieve memory-efficient training of neural networks.
The experiments on ResNet, ViT, MLP-Mixer illustrates the efficiency of the proposed method Auxiliary Activation Learning, which can significantly reduce the memory cost without sacrificing training speed.

Weakness:
Some memory-efficient algorithms are proposed in recent years, such as SM3. Some people also try to propose an efficient architecture to achieve memory-efficient training [1]. Maybe you can provide some introduction and experimental results about that.
NLP models are usually very large and memory-efficient training is also very important for them. I find you have reported the results about BERT-Large. It will be great if you can provide more results about that.

[1] Yi Tay, Mostafa Dehghani, Dara Bahri, Donald Metzler. Efficient Transformers: A Survey.

**Summary Of The Paper:**

This paper proposes a novel memory-efficient training method Auxiliary Activation Learning, which reduces the amount of data to be stored in memory without additional computations. The experimental results illustrate that Auxiliary Activation Learning can significantly reduce the memory cost on computer vision and NLP tasks without training speed reduction.

**Summary Of The Review:**

NA

---

> ### Author Response · Authors · 2022-11-13
> **Response to Reviewer zs6z**
>
> We thank the reviewer for carefully reviewing our submission and providing valuable comments. Please find our responses below:
>
> **Q1) Some memory-efficient algorithms are proposed in recent years, such as SM3. Some people also try to propose an efficient architecture to achieve memory-efficient training [1]. Maybe you can provide some introduction and experimental results about that.**
>
> **A1)** We thank the reviewer for bringing relevant work to our attention. As the reviewer suggested, we additionally implemented SM3 to compare with ASA. For BP and ASA, we used Adam optimizer. We trained BERT-Large on MRPC, which is a paraphrase identification dataset commonly used for measuring the performance of NLP models. Experimental results are summarized in Table R3.1.
>
> **Table R3.1. Test accuracy, total memory, and training time of one epoch in BERT-Large training on MRPC with 32 batch sizes.**
>
> | | BP | SM3 | ASA4 | SM3 + ASA4 |
> | --- | --- | --- | --- | --- |
> | Accuracy | 88.56 | 88.23 | 88.32 | 88.15 |
> | Total memory | 22.2 GB | 19.9 GB | 19.1 GB | 16.55 GB |
> | Training time | 1m 28s | 1m 20s | 1m 31s | 1m 22s |
>
> While our algorithm aims to reduce the amount of input activation to be stored, SM3 aims to reduce the memory required for storing states generated by accumulating previous gradients of parameters. For fair comparisons, we measured the total memory allocated to a GPU instead of the memory required to store activations. The results show that our ASA4 algorithm requires smaller memory space but takes more time to process one epoch compared to SM3. This is because the SM3 optimizer is faster than the Adam optimizer that we used for all experiments. Since the theoretical backgrounds behind these two methods are quite different, they can be combined together to obtain even larger savings. As displayed in the table above, if we apply both methods simultaneously, we can obtain the largest memory savings with minimal impact on the accuracy. Following the reviewer's suggestion, we have included SM3 in the introduction and added experimental results in Appendix G in the revised manuscript.
>
> **Q2) NLP models are usually very large and memory-efficient training is also very important for them. I find you have reported the results about BERT-Large. It will be great if you can provide more results about that.**
>
> **A2)** We thank the reviewer for this great suggestion. Taking the reviewer's advice, we experimented BERT-Large on a different dataset, MRPC, and the experimental results are summarized in the table below.
>
> **Table R3.2. Test accuracy, training memory, and training time of one epoch in BERT-Large training on MRPC with 32 batch sizes.**
>
> | | | - | ASA1 | ASA2 | ASA3 | ASA4 |
> | --- | --- | --- | --- | --- | --- | --- |
> | BP | Accuracy | 88.56 | 88.69 | 88.23 | **88.97** | 88.32 |
> | | Training memory | 10.9 GB | (1.1x) | (1.2x) | (1.3x) | (1.4x) |
> | | Training Time | 1m 28s | 1m 29s | 1m 30s | 1m 30s | 1m 31s |
> | Mesa | Accuracy | 88.3 | 88.35 | 88.51 | 88.25 | 88.23 |
> | | Training memory | (2.1x) | (2.3x) | (2.4x) | (2.4x) | ( **2.5x** ) |
> | | Training Time | 2m 8s | 2m 5s | 2m 3s | 2m 1s | 1m 55s |
>
> The table above confirms that our method can save training memory without noticeable training speed degradation on the new dataset as well. These results are now included in Table 3 of the revised manuscript.
>
> [1] Tay, Y., Dehghani, M., Bahri, D., & Metzler, D. (2020). Efficient transformers: A survey. ACM Computing Surveys (CSUR).

---

### Official Review · Reviewer_Urez · 2022-10-25

**Confidence:** 4
**Correctness:** 4
**Technical Novelty And Significance:** 2
**Empirical Novelty And Significance:** 3
**Recommendation:** 6

**Clarity, Quality, Novelty And Reproducibility:**

The overall presentation and quality is good. The method should be easy to reproduce since it is simple. Novelty somewhat limited, since some similar techniques can be founded.

**Strength And Weaknesses:**

Strengths:
- Efficient training is an important topic to study, and experiments in this paper have shown the proposed method can indeed achieve practical efficiency on training.
- The presentation of this paper is overall clear and it is easy to follow and understand.
- The proposed method is straight-forward and does not bring much overhead.

Weakness:
- The proposed technique is not essentially novel. As authors have already discussed, there are already many works trying to approximate the activations during training. Activation quantization and pruning, although argued by the authors to be slower which I do not believe, is a very simple way to achieve memory reduction.
- (Minor) The baseline on ImageNet looks weaker compared to the commonly used one, which is around 76% (1% lower approximately). It might not be negligible at the scale of ImageNet.
- (Minor) $f$ is never assumed to be non-negative (although in common practice we use a non-negative one). It is only twice differentiable and convex.

**Summary Of The Paper:**

This paper proposes a training method that saves the memory that the activations take. Instead of using the originally dense activation, the authors propose to use the auxiliary activations during the backward process. During the forward pass, the originally dense activation will still be used so the precision is still high. The authors discuss two ways to construct the auxiliary activation: the residual and the sign. Experiments on image classification and NLP tasks show that the training time and the memory can be saved, while the performance on downstream task is not sacrificed too much.

**Summary Of The Review:**

This work proposes an effective way to efficiently train models with less memory cost, although the novelty is not outstanding in its current shape.

---

> ### Author Response · Authors · 2022-11-13
> **Response to Reviewer Urez**
>
> [1] Chen, T., Xu, B., Zhang, C., & Guestrin, C. (2016). Training deep nets with sublinear memory cost. arXiv preprint arXiv:1604.06174.
>
> [2] Chakrabarti, A., & Moseley, B. (2019). Backprop with approximate activations for memory-efficient network training. Advances in Neural Information Processing Systems, 32.
>
> [3] Chen, J., Zheng, L., Yao, Z., Wang, D., Stoica, I., Mahoney, M., & Gonzalez, J. (2021, July). Actnn: Reducing training memory footprint via 2-bit activation compressed training. In International Conference on Machine Learning (pp. 1803-1813). PMLR.
>
> [4] Evans, R. D., & Aamodt, T. (2021). AC-GC: Lossy Activation Compression with Guaranteed Convergence. Advances in Neural Information Processing Systems, 34, 27434-27448.
>
> [5] Liu, X., Zheng, L., Wang, D., Cen, Y., Chen, W., Han, X., … & Cheung, A. (2022, June). GACT: Activation compressed training for generic network architectures. In International Conference on Machine Learning (pp. 14139-14152). PMLR.
>
> [6] Pan, Z., Chen, P., He, H., Liu, J., Cai, J., & Zhuang, B. (2021). Mesa: A Memory-saving Training Framework for Transformers. arXiv preprint arXiv:2111.11124.
>
> [7] Gomez, A. N., Ren, M., Urtasun, R., & Grosse, R. B. (2017). The reversible residual network: Backpropagation without storing activations. Advances in neural information processing systems, 30.
>
> [8] Sander, M. E., Ablin, P., Blondel, M., & Peyré, G. (2021, July). Momentum residual neural networks. In International Conference on Machine Learning (pp. 9276-9287). PMLR.
>
> [9] Huo, Z., Gu, B., & Huang, H. (2018, July). Decoupled parallel backpropagation with convergence guarantee. In International Conference on Machine Learning (pp. 2098-2106). PMLR.
>
> [10] Huo, Z., Gu, B., & Huang, H. (2018). Training neural networks using features replay. Advances in Neural Information Processing Systems, 31.
>
> [11] Belilovsky, E., Eickenberg, M., & Oyallon, E. (2020, November). Decoupled greedy learning of cnns. In International Conference on Machine Learning (pp. 736-745). PMLR.

---

> ### Author Response · Authors · 2022-11-13
> **Response to Reviewer Urez**
>
> **Q3) (Minor) f is never assumed to be non-negative (although in common practice we use a non-negative one). It is only twice differentiable and convex.**
>
> **A3)** We thank the reviewer for valuable feedback. First of all, we apologize that there was a mistake in the initial draft. We initially assumed that
>
> $f$ **is convex, differentiable and Lipschitz continuous**
>
> , so that Theorem 1 can be proven using the quadratic upper bound as shown in Appendix A.
>
> **(quadratic upper bound)** $f(y) \leq f(x) + \nabla f(x)^{T}(y-x) + \frac{L}{2}\lVert y-x \rVert_{2}^{2}$
>
> However, the inequality above actually holds for non-convex functions as well. Therefore, in the revised manuscript, we have corrected our assumption as follows:
>
> **The gradient of $f(x)$ is Lipschitz continuous**
>
> Under the assumption above, we can induce the quadratic upper bound even if is a non-convex function. Detailed proof can be found at the link below:
>
> **(proof)** https://drive.google.com/file/d/10L0DDOrD0IBoP8Zd7ZiaHZRqpBmbqmnc/view?usp=share_link
>
> Please note that the constraint of the gradient of being Lipschitz continuous has been widely used in prior work on analyzing deep neural network training and reducing training memory [3-5, 9-11]. Therefore, we believe that we are taking an identical approach to prior work to prove the effectiveness of the proposed algorithm; we first theoretically show that it works under some constraints, which were also employed in prior work, and experimentally demonstrate its performance using various models.
>
> We also agree with the reviewer that is usually a non-negative function. However, since we proved Theorem 1 without using this constraint, it will hold even if is non-negative. We may find more interesting aspects utilizing this property of , which we will continue to study in future work.

---

> ### Author Response · Authors · 2022-11-13
> **Response to Reviewer Urez**
>
> **Q2) (Minor) The baseline on ImageNet looks weaker compared to the commonly used one, which is around 76% (1% lower approximately). It might not be negligible at the scale of ImageNet.**
>
> **A2)** We thank the reviewer for catching this. In experiments, we trained ResNet-50 with a 0.1 initial learning rate, an SGD optimizer, and a step scheduler during 90 epochs. The accuracy would be increased by using warm-up steps and additional hyperparameter tuning. Unfortunately, exploring these options would take a long time. To address the reviewer's concern, we will continue these experiments and update experimental results in the final version if accepted.

---

> > ### Author Response · Authors · 2022-12-03
> > **Response to Reviewer Urez**
> >
> > **Q2-1)** **(Minor) The baseline on ImageNet looks weaker compared to the commonly used one, which is around 76% (1% lower approximately). It might not be negligible at the scale of ImageNet.**
> >
> > **A2-1)** In previous experiments, we employed the initial learning rate of 0.1 and reduced it by a factor of 0.1 every 30 epochs during 90 epochs in previous experiments. To address the reviewer's concern,we applied four warm-up steps and cosine annealing with a 5e-2 initial learning rate during 90 epochs. As a result, the test accuracy of the baseline model trained using BP has been increased to 76.01%. We performed the same experiments using this updated baseline model, and experimental results are summarized in the table below (Table R2.5), along with the original results reported in the initial submission (Table R2.4). The results confirm that our ARA scheme still achieves competitive performance close to BP. We will update the experimental results in the final version of the manuscript if accepted.
> >
> > **Table R2.4. Test accuracy, training memory, and training time of one epoch in ResNet-50 training on ImageNet with 512 batch sizes (previous baseline).**
> >
> > | | - | ARA (2,2,2,2) | ARA (3,4,2,2) | ARA (3,4,4,2) | ARA (3,4,6,2) |
> > | --- | --- | --- | --- | --- | --- |
> > | BP | 75.22 | 74.96 | 74.92 | 74.85 | 75.52 |
> > | GCP | 75.22 | 74.96 | 74.92 | 74.85 | 74.52 |
> > | ActNN | 75.03 | 74.93 | 74.87 | 74.65 | 74.5 |
> >
> > **Table R2.5. Test accuracy, training memory, and training time of one epoch in ResNet-50 training on ImageNet with 512 batch sizes (updated baseline).**
> >
> > | | - | ARA (2,2,2,2) | ARA (3,4,2,2) | ARA (3,4,4,2) | ARA (3,4,6,2) |
> > | --- | --- | --- | --- | --- | --- |
> > | BP | 76.01 | 75.97 | 75.89 | 75.62 | 75.23 |
> > | GCP | 76.01 | 75.97 | 75.89 | 75.62 | 75.23 |
> > | ActNN | 75.96 | 75.93 | 75.67 | 75.51 | 75.12 |

---

> ### Author Response · Authors · 2022-11-13
> **Response to Reviewer Urez**
>
> We thank the reviewer for carefully reviewing our submission and providing valuable comments. Please find our responses below:
>
> **Q1) The proposed technique is not essentially novel. As authors have already discussed, there are already many works trying to approximate the activations during training. Activation quantization and pruning, although argued by the authors to be slower which I do not believe, is a very simple way to achieve memory reduction.**
>
> **A1)** We thank the reviewer for pointing out a crucial point. We agree that our algorithm is in line with other memory-saving techniques that approximate actual activation. However, we would like to emphasize that our work is meaningful in that it suggests an entirely new way to approximate activation. More specifically, prior approaches reduce training memory through gradient checkpointing [1], quantizing activation [2-6], or using reversible neural network architectures [7-8], whereas our algorithm replaces actual activation with auxiliary activation. As a result, the proposed algorithm exhibits three advantages: i) minimal impact on the training speed, ii) high degree of applicability, and iii) being orthogonal to other techniques.
>
> First, prior methods degrade the training speed because they require additional computation. For instance, GCP and reversible networks require recomputing activations during backward propagation. ActNN and Mesa need to process additional operations such as computing min/max values, stochastic rounding, tensor reshaping, and Hadamard product for quantization/dequantization. In addition, ActNN additionally needs a greedy algorithm to allocate bits for mixed-precision quantization. In our experiments, we used the authors' open-source implementation based on CUDA to maximize speed for ActNN [3] and Mesa [6], a large training speed reduction was still observed, as shown in Tables R2.1 and R2.2. This trend is also consistent with the observation provided in previous studies; for instance, the authors in [3, 4, 6] indicate that the training speed could be significantly reduced when using their methods. Contrarily, our ARA does not require this additional operation for training, resulting in substantially faster training than the other algorithms. Although ASA needs to pack auxiliary sign activation, it is implemented using a simple bitwise operation, resulting in a substantially smaller impact on the training speed.
>
> Second, some prior studies report significant memory savings by modifying neural network architectures into reversible structures. However, this is not applicable to other types of neural networks since it is not always straightforward to transform a well-performing non-reversible model into a reversible one. On the other hand, our algorithm is applicable to a wide range of neural network models. ARA can be used on any model with residual connections, and ASA can be applied to all networks without restriction.
>
> Finally, the theoretical background of our algorithm is different from other memory-saving techniques. As a result, our algorithm can be combined with other methods for even larger memory savings. For instance, our ARA and ASA could be successfully combined with GCP, ActNN, and Mesa to reduce training memory further, as demonstrated in the experimental results in the manuscript.
>
> **Table R2.1. Test accuracy, training memory, and training time of one epoch in ResNet-152 training on ImageNet with 512 batch sizes.**
>
> | | BP | ARA (3,4,2,2) | GCP | ActNN |
> | --- | --- | --- | --- | --- |
> | Accuracy | 77.38 | 77.41 | 77.38 | 77.15 |
> | Training memory | 90.2 GB | 74.5 GB (x1.21) | 74.4 GB (x1.21) | 74.5 GB (x1.21) |
> | Training time | 35m 37s | 36m 24s | 45m 21s | 47m 39s |
>
> **Table R2.2. Test accuracy, training memory, and training time of one epoch in ViT-Large training on CIFAR-100 with 128 batch sizes.**
>
> | | BP | ASA3 | Mesa |
> | --- | --- | --- | --- |
> | Accuracy | 92.93 | 92.97 | 92.91 |
> | Training memory | 48 GB | 33.1 GB (x1.5) | 31.8 GB (x1.5) |
> | Training time | 2m 46s | 2m 51s | 3m 12s |

---

> > ### Comment · Reviewer_Urez · 2022-11-22
> > **Thank you for your response**
> >
> > Thanks to the authors for addressing the concerns I have raised. From the table, one could indeed observe that the time saving looks significant. My final (minor, not necessary) question would be: maybe the authors could provide a comparison against direct low-precision training (e.g. 16-bit, 8-bit) to further demonstrate the quality of the auxiliary activation.

---

> > > ### Author Response · Authors · 2022-12-03
> > > **Response to Reviewer Urez**
> > >
> > > **Q1-1)** **My final (minor, not necessary) question would be: maybe the authors could provide a comparison against direct low-precision training (e.g. 16-bit, 8-bit) to further demonstrate the quality of the auxiliary activation.**
> > >
> > > **A1-1)** We thank the reviewer for this insightful suggestion. Following the reviewer's suggestion, we performed additional experiments using quantized activations. For fair comparisons, we quantized the activations of the convolutional layers in which we applied ARA into 16 and 8 bits to compare their quality, and we employed the data formats developed for low-precision training (FP16 [1] and FP8 [2]). Experimental results are shown below.
> > >
> > > **Table R2.3. Test accuracy, training memory, and training time of one epoch in ResNet-50 training on CIFAR-100 with 128 batch sizes. The sharing stride of ARA is set to (3,4,2,2).**
> > >
> > > | | FP32 (BP) | FP16 | FP8 | ARA |
> > > | --- | --- | --- | --- | --- |
> > > | Accuracy | 76.83 | 76.79 | 76.72 | 76.78 |
> > > | Training memory | 3.30 GB | 3.24 GB | 3.09 GB | 2.71 GB |
> > > | Training time | 51.54s | 53.76s | 53.94s | 53.49s |
> > >
> > > The table above shows that 16-bit and 8-bit quantized activations exhibit similar training performance to BP. This is an expected result since ActNN [3] already adopts 2-bit activations and achieves high accuracy. For instance, Table 2 of our submission shows that ActNN achieves 75.03% accuracy for ResNet-50 training on ImageNet, closely matching that of BP (75.22%). Since our ARA scheme also exhibits competitive performance close to BP, we can conclude that the quality of auxiliary activation is similar to that of full-precision and quantized activations. However, as discussed in our previous response, our ARA scheme is meaningful in that it is based on different theoretical backgrounds and hence can be successfully combined with other memory-saving techniques, including activation quantization such as ActNN, for larger memory savings without accuracy degradation.
> > >
> > > [1] Paulius Micikevicius, Sharan Narang, Jonah Alben, Gregory F. Diamos, Erich Elsen, David García, Boris Ginsburg, Michael Houston, Oleksii Kuchaiev, Ganesh Venkatesh, & Hao Wu. Mixed Precision Training, _ICLR 2018, Vancouver, BC, Canada, April 30 - May 3, 2018, Conference Track Proceedings_.
> > >
> > > [2] Wang, N., Choi, J., Brand, D., Chen, C. Y., & Gopalakrishnan, K. Training deep neural networks with 8-bit floating point numbers. _Advances in neural information processing systems_, _31_.
> > >
> > > [3] Chen, J., Zheng, L., Yao, Z., Wang, D., Stoica, I., Mahoney, M., & Gonzalez, J. (2021, July). Actnn: Reducing training memory footprint via 2-bit activation compressed training. In International Conference on Machine Learning (pp. 1803-1813). PMLR.

---

### Official Review · Reviewer_91fV · 2022-10-26

**Confidence:** 4
**Correctness:** 3
**Technical Novelty And Significance:** 4
**Empirical Novelty And Significance:** Not applicable
**Recommendation:** 8

**Clarity, Quality, Novelty And Reproducibility:**

The paper is well written, and easy to follow. It proposes a completely novel approach that makes it possible to compress the activations by 1.1x to 1.3x depending on the scenario with no noticeable impact on accuracy and negligible impact on training time, The authors have released the code needed to reproduce their results, though I haven't had a chance to use it.


**Strength And Weaknesses:**

Strengths:
* The approach is compelling and completely novel as far as I know. It is effective on its own and can be combined with other existing approaches to save more memory.
* The learning indicator criteria is sound under reasonable assumptions, and used effectively to develop two schemes, ARA and ASA.
* The empirical evaluation of the scheme is comprehensive and demonstrates that the approach is applicable to a wide range of DNN architecture.

Weaknesses:
* There is a minor error in Theorem 1: $\eta$ must be strictly positive for the loss function to converge
* The convexity assumption under which the learning indicator criteria was developed often does not hold in many cases. That said, most of our theoretical understanding of deep learning suffers from the same limitation, and empirical proof is used to prove that this approach works in practice.
* Table 1 doesn't list the training speed and memory reduction like the other tables do. It would have been great to be able to compare ASA and ARA side by side. I am especially curious to see whether there is a correlation between training speed and mean value of the learning indicator.
* In table 2, the speed and accuracy of ARA is compared against that of GCP and ActNN. The comparison would be more fair if there was a way to decrease the compression rate achieved by GCP and ActNN to be in line with these of ARA. The same applies to table 3.
* Table 4 and 5 propose to use the memory savings to increase the batch size, but this isn't very effective with less than 5% savings in the best case. I wonder if the authors could instead try to increase the width/depth the the neural networks to improve the accuracy of the predictions instead, which I suspect is what most people will use the memory savings for.

**Summary Of The Paper:**

Training deep learning models is becoming increasingly challenging: there is simply not enough memory to store the activations generated by the forward pass until the corresponding gradients are computed. The paper proposes to split each activations in two tensors, $o$ and $a$. The $o$ tensor can be discarded quickly, while the $a$ tensor needs to be preserved in memory to compute the gradients.  The paper develops a theoretical criteria, the learning indicator, under which this partitioning scheme is guaranteed to succeed (subject to some reasonable assumptions). Based on this criteria, the authors design two schemes to partition the activations, auxiliary residual activation (ARA) and auxiliary sign activations (ASA), and empirically demonstrate their effectiveness on a comprehensive set of models.


**Summary Of The Review:**

 The memory bottleneck is an acute problem that impacts most of the deep learning community This paper provides a novel solution that alleviates this problem. Furthermore, this new approach can be effectively combined with existing solutions such as rematerialization to save more memory.

---

> ### Author Response · Authors · 2022-11-13
> **Title: Response to Reviewer 91fV**
>
> [1] Chen, J., Zheng, L., Yao, Z., Wang, D., Stoica, I., Mahoney, M., & Gonzalez, J. (2021, July). Actnn: Reducing training memory footprint via 2-bit activation compressed training. In International Conference on Machine Learning (pp. 1803-1813). PMLR.
>
> [2] Evans, R. D., & Aamodt, T. (2021). AC-GC: Lossy Activation Compression with Guaranteed Convergence. Advances in Neural Information Processing Systems, 34, 27434-27448.
>
> [3] Liu, X., Zheng, L., Wang, D., Cen, Y., Chen, W., Han, X., ... & Cheung, A. (2022, June). GACT: Activation compressed training for generic network architectures. In International Conference on Machine Learning (pp. 14139-14152). PMLR.
>
> [4] Huo, Z., Gu, B., & Huang, H. (2018, July). Decoupled parallel backpropagation with convergence guarantee. In International Conference on Machine Learning (pp. 2098-2106). PMLR.
>
> [5] Huo, Z., Gu, B., & Huang, H. (2018). Training neural networks using features replay. Advances in Neural Information Processing Systems, 31.
>
> [6] Belilovsky, E., Eickenberg, M., & Oyallon, E. (2020, November). Decoupled greedy learning of cnns. In International Conference on Machine Learning (pp. 736-745). PMLR.

---

> > ### Comment · Reviewer_91fV · 2022-11-21
> > **Response to authors**
> >
> > I thank the authors for their time and careful responses to my questions, which address my concerns.

---

> ### Author Response · Authors · 2022-11-13
> **Response to Reviewer 91fV**
>
> **Q5) Table 4 and 5 propose to use the memory savings to increase the batch size, but this isn't very effective with less than 5% savings in the best case. I wonder if the authors could instead try to increase the width/depth the neural networks to improve the accuracy of the predictions instead, which I suspect is what most people will use the memory savings for.**
>
> **A5)** This is a very valid point. Following the reviewer's suggestion, we scaled ResNet-152 and BERT-Large using the same batch size to confirm the effectiveness of our algorithm. For ResNet-152, we fixed other parameters and increased the number of layers or the width of the bottleneck block following the scheme in [1, 3]. We also scaled BERT-Large by increasing the number of transformer blocks or the hidden size following the scheme in [3]. The results are as follows:
>
> **Table R1.4. Largest model based on ResNet-152 that can be trained using a single GPU with 24GB memory (depth: number of layers, width: width of the first bottleneck block).**
>
> || BP | ARA (3,4,6,2) | ActNN | ActNN + ARA (3,4,6,2) |
> | --- | --- | --- | --- | --- |
> | Depth | 146 | 165 | 622 | 718 |
> | Width | 62 | 76 | 214 | 238 |
>
> **Table R1.5. Largest model based on BERT-Large that can be trained using a single GPU with 24GB memory (depth: number of transformer blocks, width: hidden size).**
>
> | | BP | ASA4 | Mesa | Mesa + ASA4 |
> | --- | --- | --- | --- | --- |
> | Depth | 50 | 60 | 64 | 70 |
> | Width | 1600 | 1728 | 1792 | 1856 |
>
> Using ARA, we can train 13% deeper or 22% wider networks compared to using BP only. Similarly, additionally applying ARA allows for using 15% deeper or 11% wider models compared to using ActNN only. For BERT-Large, we can train 20% deeper or 8% wider models by using ASA. Additionally applying ASA allows for training 9% deeper or 4% wider models compared to using Mesa only. These results are summarized in Section 4.3 of the revised manuscript.

---

> ### Author Response · Authors · 2022-11-13
> **Response to Reviewer 91fV**
>
> **Q4) In table 2, the speed and accuracy of ARA is compared against that of GCP and ActNN. The comparison would be more fair if there was a way to decrease the compression rate achieved by GCP and ActNN to be in line with these of ARA. The same applies to table 3.**
>
> **A4)** We thank the reviewer for this great suggestion. We can decrease the compression rate of GCP, ActNN, and Mesa by selectively applying these methods to some layers rather than the entire network. In experiments, we applied GCP to the last Conv-BN layer of residual blocks in Fig. 3a because the dimension of input activation in the last batch normalization layer is equal to that of the ARA-Conv layer, which is not stored in memory when applying ARA. The input activation of the batch normalization layer is not stored in the Conv-BN layer, but it is recomputed from the input activation of the convolutional layer during backward propagation. Through this, an identical amount of memory saving can be achieved by using GCP. For ActNN, we applied it to the first convolutional layer of residual blocks as we did for ARA in Fig. 3a, which resulted in an identical compression rate. While our ASA stores 1-bit auxiliary sign activation, Mesa stores 8-bit compressed activation. Therefore, we cannot achieve the same memory saving by applying Mesa in the same way as ASA. To overcome this issue, we additionally apply Mesa to the batch normalization and layer normalization layers to obtain the same compression rate to ASA. Tables R1.2 and R1.3 show that ARA and ASA train the networks faster than other memory-saving algorithms with identical compression rates as they do not require additional costly computations. These results are now included in a new section (Appendix D) in the revised manuscript.
>
> **Table R1.2. Test accuracy, training memory, compression rate (bracketed), and training time of one epoch in ResNet-152 training on ImageNet with 512 batch sizes.**
>
> | | BP | ARA (3,4,2,2) | GCP | ActNN |
> | --- | --- | --- | --- | --- |
> | Accuracy | 77.38 | 77.41 | 77.38 | 77.15 |
> | Training memory | 90.2 GB | 74.5 GB (x1.21) | 74.4 GB (x1.21) | 74.5 GB (x1.21) |
> | Training time | 35m 37s | 36m 24s | 45m 21s | 47m 39s |
>
> **Table R1.3. Test accuracy, training memory, compression rate (bracketed), and training time of one epoch in ViT-Large training on CIFAR-100 with 512 batch sizes.**
>
> | | BP | ASA3 | Mesa |
> | --- | --- | --- | --- |
> | Accuracy | 92.93 | 92.97 | 92.91 |
> | Training memory | 48 GB | 33.1 GB (x1.5) | 31.8 GB (x1.5) |
> | Training time | 2m 46s | 2m 51s | 3m 12s |

---

> ### Author Response · Authors · 2022-11-13
> **Response to Reviewer 91fV**
>
> **Q3) Table 1 doesn't list the training speed and memory reduction like the other tables do. It would have been great to be able to compare ASA and ARA side by side. I am especially curious to see whether there is a correlation between training speed and mean value of the learning indicator.**
>
> **A3)** We thank the reviewer for valuable feedback. Following the reviewer's suggestion, we have added a new section in the appendix (Appendix B) to discuss this point in more detail. The table below (Table 5 in Appendix B) compares ARA and ASA including training speed, memory reduction, and the mean value of the learning indicator.
>
> **Table R1.1. Test accuracy, training memory, training time of one epoch, and mean values of learning indicator in ResNet-18 training.**
>
> | Dataset | Metric | BP | ARA (2, 2, 2, 2) | ASA (2, 2, 2, 2) |
> | --- | --- | --- | --- | --- |
> | CIFAR-10 | Accuracy | 94.77 | 94.81 | 94.76 |
> | | Training memory | 605 MB | 546 MB | 549 MB |
> | | Training time | 20.83s | 21.30s | 21.46s |
> | | Learning Indicator | - | 0.69 | 0.03 |
> | CIFAR-100 | Accuracy | 75.81 | 75.49 | 75.31 |
> | | Training memory | 605 MB | 546 MB | 549 MB |
> | | Training time | 20.86s | 21.32s | 21.52s |
> | | Learning Indicator | - | 0.71 | 0.03 |
> | Tiny-ImageNet | Accuracy | 58.43 | 58.46 | 57.01 |
> | | Training memory | 1211 MB | 1091 MB | 1097 MB |
> | | Training time | 36.82s | 37.54s | 37.94s |
> | | Learning Indicator | - | 0.7 | 0.03 |
>
> The training memory of ASA is slightly larger than that of ARA since ASA has to obtain and store 1-bit auxiliary sign activation, while ARA does not need to store actual activation and just reuses auxiliary activation.
>
> We also measured the mean values of the learning indicator. The mean values of the learning indicator of ARA are significantly higher than those of ASA. Therefore, from Theorem 1, we can expect that the loss of the network would converge better by using ARA compared to ASA. This is consistent with experimental results in the table above; ARA achieves higher accuracy than ASA in all experiments. In addition, both train and validation losses converge better when using ARA, as shown in the learning curves below.
>
> **(Train loss)**
>
> [https://drive.google.com/file/d/1QUXMSxI5bU1zJNlxCKWwdVYjm5\_--YbY/view?usp=share\_link](https://drive.google.com/file/d/1QUXMSxI5bU1zJNlxCKWwdVYjm5_--YbY/view?usp=share_link)
>
> **(Validation loss)**
>
> [https://drive.google.com/file/d/1atKDKrm8525R00aVnnzuhIUvIuE0xRw2/view?usp=share\_link](https://drive.google.com/file/d/1atKDKrm8525R00aVnnzuhIUvIuE0xRw2/view?usp=share_link)
>
> The processing time for one epoch of ASA is slightly larger than that of ARA since ASA needs to pack 1-bit auxiliary sign activation, whereas ARA employs auxiliary residual activation as is. These results and discussions have been included in Appendix B in the revised manuscript.

---

> ### Author Response · Authors · 2022-11-13
> **Response to Reviewer 91fV**
>
> **Q2) The convexity assumption under which the learning indicator criteria was developed often does not hold in many cases. That said, most of our theoretical understanding of deep learning suffers from the same limitation, and empirical proof is used to prove that this approach works in practice.**
>
> **A2)** We thank the reviewer for pointing out an important point. First of all, we apologize that there was a mistake in the initial draft. We initially assumed that
>
> $f$ **is convex, differentiable and Lipschitz continuous**
>
> , so that Theorem 1 can be proven using the quadratic upper bound as shown in Appendix A.
>
> **(quadratic upper bound)** $f(y) \leq f(x) + \nabla f(x)^{T}(y-x) + \frac{L}{2}\lVert y-x \rVert_{2}^{2}$
>
> However, the inequality above actually holds for non-convex functions as well. Therefore, in the revised manuscript, we have corrected our assumption as follows:
>
> **The gradient of $f(x)$ is Lipschitz continuous**
>
> Under the assumption above, we can induce the quadratic upper bound even if is a non-convex function. Detailed proof can be found at the link below:
>
> **(proof)** https://drive.google.com/file/d/10L0DDOrD0IBoP8Zd7ZiaHZRqpBmbqmnc/view?usp=share_link
>
> Please note that the constraint of the gradient of being Lipschitz continuous has been widely used in prior work on analyzing deep neural network training and reducing training memory [1-6]. Therefore, we believe that we are taking an identical approach to prior work to prove the effectiveness of the proposed algorithm; we first theoretically show that it works under some constraints, which were also employed in prior work, and experimentally demonstrate its performance using various models.

---

> ### Author Response · Authors · 2022-11-13
> **Response to Reviewer 91fV**
>
> We thank the reviewer for carefully reviewing our submission and providing valuable comments. Please find our responses below:
>
> **Q1)**  **There is a minor error in Theorem 1: must be strictly positive for the loss function to converge**
>
> **A1)** We thank the reviewer for catching this and apologize for our mistake. The inequality in Theorem 1 has been corrected to $0<\eta\leq\frac{1}{L}$ in the revised manuscript.

---

### Public Comment · ~Yucheng_Ding1 · 2023-09-25
**A little question about the paper.**

Dear Authors,

I'm very interested in your solid paper, but I have a little question: does y_l (the input of an activation layer, in Equation (6)) need to be stored in the memory to calculate the gradient \phi'(y_l)?

Thank you very much in any case.

Best regards,

Yucheng Ding

---

> ### Author Response · Authors · 2023-10-04
> **Dear Yucheng Ding**
>
> Thank you for carefully reading our paper.
> Your comment is right. As indicated in equation (6), our algorithm does require storing the input $y_{l}$ of the activation layer.
> However, the primary goal of our algorithm is to save memory on the inputs of the linear or convolutional layers, not the activation layer.
>
> That is, in the backpropagation algorithm (i.e., equations (1)-(3)), both $h_{l}$ and $y_{l}$ need to be stored in order to compute $\Delta W_{l+1}$ and $\delta_{l}$. On the other hands, our AAL (i.e., equations (4)-(6)) need to store  $a_{l}$ and $y_{l}$ for calculating them. Here, by sharing ​$a_{l}$ across multiple layers through residual connections (ARA) or using the sign bit (ASA) as demonstrated in the paper, we can save the memory that would be otherwise allocated for $h_{l}$.

---

### Decision · Program_Chairs · 2023-01-20

**Decision:**

Accept: poster

**Justification For Why Not Higher Score:**

There seems to be a good amount of novelty here, and the proposed method does seem effective, but the gains are somewhat minimal. A good paper for exploring new and unusual methods, but probably will not see wide adoption.

**Justification For Why Not Lower Score:**

All reviewers recommended acceptance.

**Metareview: Summary, Strengths And Weaknesses:**

This paper proposes an alternative to backpropagation that enables avoiding caching the forward-pass activations for reuse during the backwards pass. In particular, the method introduces an auxiliary activation during the forward pass that is added to each layer's input and is used alone during the backwards pass. The auxiliary activation can be shared across layers and/or can be small relative to the activations that would be stored during backprop. The approach provides some modest improvements in terms of memory savings without sacrificing performance or speed, and can be applied on top of existing methods for saving memory. The paper includes extensive experiments in a variety of settings. Reviewers felt the paper was clear and that the method was novel and well-justified. All reviewers recommended acceptance.

**Note From Pc:**

if the above contains the word "oral" or "spotlight" please see: "oral" presentation means -> notable-top-5% and "spotlight" means -> notable-top-25%. As stated in our emails, we are disassociating presentation type from AC recommendations